# Wisp1 is a circulating factor that stimulates proliferation of adult mouse and human beta cells

Rebeca Fernandez-Ruiz[1,2,3], Ainhoa García-Alamán[1,2], Yaiza Esteban[1,2], Joan Mir-Coll[1,3], Berta Serra-Navarro[1,3], Marta Fontcuberta-PiSunyer[1], Christophe Broca[4], Mathieu Armanet[4], Anne Wojtusciszyn[4,5], Vardit Kram[6], Marian F. Young[6], Josep Vidal[1,2,7], Ramon Gomis[1,2,3,8] & Rosa Gasa [1,2 ✉]

Expanding the mass of pancreatic insulin-producing beta cells through re-activation of beta cell replication has been proposed as a therapy to prevent or delay the appearance of diabetes. Pancreatic beta cells exhibit an age-dependent decrease in their proliferative activity, partly related to changes in the systemic environment. Here we report the identification of CCN4/Wisp1 as a circulating factor more abundant in pre-weaning than in adult mice. We show that Wisp1 promotes endogenous and transplanted adult beta cell proliferation in vivo. We validate these findings using isolated mouse and human islets and find that the beta cell trophic effect of Wisp1 is dependent on Akt signaling. In summary, our study reveals the role of Wisp1 as an inducer of beta cell replication, supporting the idea that the use of young blood factors may be a useful strategy to expand adult beta cell mass.

[1] August Pi i Sunyer Biomedical Research Institute (IDIBAPS), Rosselló 149-153, 08036 Barcelona, Spain. [2] Centro de Investigación Biomédica en Red de Diabetes y Enfermedades Metabólicas Asociadas (CIBERDEM), Barcelona, Spain. [3] University of Barcelona, Barcelona, Spain. [4] CHU Montpellier, Laboratory of Cell Therapy for Diabetes (LTCD), Hospital St-Eloi, Montpellier, France. [5] Service of Endocrinology, Diabetes and Metabolism, Lausanne University Hospital, Lausanne, Switzerland. [6] Molecular Biology of Bones and Teeth Section, NIDCR, National Institutes of Health, Bethesda, MD, USA. [7] Department of Endocrinology and Nutrition, Hospital Clinic of Barcelona, Barcelona, Spain. [8] Universitat Oberta de Catalunya (UOC), Barcelona, Spain. ✉email: rgasa@clinic.cat

Loss of functional beta cell mass is a central feature in the pathogenesis of diabetes, and restoration of the beta cell mass reservoir through replication of remaining beta cells has been envisioned as a potential therapeutic strategy to treat this disease. This has fueled a growing interest in deciphering the molecular pathways that direct beta cell proliferation and increasing efforts to identify exogenous factors that can be used to promote beta cell replication in situ and in vitro.

The main mechanism by which beta cells expand themselves is through replication, but their ability to proliferate declines with age, and, therefore, adult beta cells are considered resistant to different proliferative stimuli[1]. In the mouse, beta cell replication is high in the late embryonic and perinatal periods and progressively diminishes until reaching low levels after weaning. Several studies indicate that intrinsic changes in beta cells, such as an age-dependent increase in p16[INK4A], a cyclin-dependent kinase inhibitor, restrict the proliferation of both old mouse and human beta cells[2]. Likewise, loss of islet responsiveness to platelet-derived growth factor (PDGF) signaling due to down-regulation of the expression of PDGF receptors has been associated to the decline in beta cell proliferation after birth[3].

As it has been shown for other cell types, the age-dependent decrease in beta cell proliferation not only depends on intrinsic molecular changes, but it is also the consequence of modifications of the systemic environment that cells are exposed to[4–6]. In this regard, parabiosis experiments in mice demonstrated that old beta cells regain the ability to proliferate when they are exposed systemically to blood from young mice[7]. Despite the growing body of research on molecules that can act as beta cell trophic factors in vitro and/or in vivo[8–11], the nature of the factor/s present in young blood that function as stimulators of beta cell growth remains elusive.

Matricellular proteins are nonstructural proteins present in the extracellular cellular matrix that, rather than having stable structural roles, tend to exhibit rapid turnover and play dynamic roles in multiple cellular processes, including cell proliferation, programmed cell death, extracellular matrix production or cellular migration[12]. These proteins exhibit a largely context-specific mode of action based on their ability to interact with multiple partners, including major structural elements in the matrix, specific cell surface receptors and growth factors. Among these matricellular proteins, there is the CCN protein family[13]. There are six CCN proteins, designated CCN1 to CCN6, which share a modular structure comprised of a signal peptide followed by three domains with homology to insulin-like growth factor binding protein, von Willebrand type C and thrombospondin type 1 repeat, plus a fourth domain which contains a cysteine-knot motif[14]. This modular structure underlies the pleiotropic functions and cell-specific behavior of this family of proteins, as CCN proteins can use different cell-surface receptors in different cell types to interact with multiple partners present in the surrounding extracellular matrix[15]. Two members of the CCN family, CCN1/Cyr61 and CCN2/CTGF, are expressed in the pancreas where they have been associated with proliferative processes. On one hand, upregulation of CCN1/Cyr61 in pancreatic carcinoma cells alters expression of genes encoding MAPK/ERK and PI3K and stimulates cell division in response to growth factors[16]. On the other hand, CCN2/CTGF is expressed in mouse embryonic beta cells and re-expressed in adult beta cells during periods of expansion such as pregnancy or after high fat feeding[17].

CCN4/Wisp1 (Wnt-1 inducible signaling protein 1) was first identified as a target of the Wnt pathway in mammary epithelial cells[18]. Wisp1 was subsequently found to be expressed in multiple sites within the body and to be involved in a broad spectrum of biological functions and pathological processes. Thus, Wisp1 participates in development, tumorigenesis and tissue repair through regulation of apoptosis, adhesion, migration, differentiation, and proliferation processes[19–21]. In the musculoskeletal system, Wisp1 is produced by osteoblasts and their precursors and has been shown to be pivotal for osteoblastic differentiation during bone development[22], and to assist with fracture repair[23] and growth plate cartilage injury[24]. Wisp1 has also been described as a pro-inflammatory adipokine[25]. Its expression in adipose tissue is lowered by diet-induced weight loss in human subjects, whilst it augments in response to high fat diet feeding in mice[25,26]. Furthermore, circulating levels of WISP1 and WISP1 expression in visceral adipose tissue were found increased in obesity, irrespective of type 2 diabetes status, and associated with insulin resistance and adipose tissue inflammation[27,28].

In the present study, we aimed at identifying blood factors present in pre-weaning stages that may contribute to high rates of beta cell proliferation during this period. Using antibody arrays, we identified Wisp1 as a protein enriched in serum from lactating as compared to adult mice. Because the role of Wisp1 in beta cell physiology has not been previously addressed, here we sought to investigate the potential of Wisp1 as a beta cell trophic factor.

## Results

**Adult beta cells exhibit enhanced proliferation when transplanted into pre-weaning mice.** We first examined whether a young environment could increase the proliferation of adult beta cells. To this aim, we performed syngeneic transplants of islets isolated from 20-week-old (20wo) C57BL6/J mice into the anterior chamber of the eye (ACE) of adult or postnatal day 16 (p16) C57BL6/J recipients. We found that, 12-days post-transplantation, the proportion of proliferating beta cells, both counting cells positive for the proliferation marker ki67 (marks cells engaged in the cell cycle) or for the marker of cellular mitosis pHH3 (phosphorylated histone H3), were higher in the grafts implanted in p16 relative to 20wo recipients (Fig. 1a–c). To overrule age-associated changes in graft vascularization that could have influenced this result, we performed immunostaining against CD31/PECAM-1, a vascular marker, and in vivo confocal imaging of blood vessels using rhodamine dextran. As illustrated in Supplementary Fig. 1 and Supplementary Movies 1–6, there were no obvious differences in vascularization between p16 and adult eye grafts. Next, we transplanted human islets isolated from adult individuals (55 and 56 years of age) into the ACE of immuno-compromised p16 or 20wo NSG-SCID mice. Similar to their mouse counterparts, human beta cells proliferated more (as indicated by ki67 and pHH3 staining) when transplanted into young relative to adult mouse recipients (Fig. 1d–f). Together, these experiments support the notion that the young circulatory systemic environment stimulates the proliferation of adult mouse and human beta cells.

**Identification of Wisp1 as a factor enriched in pre-weaning mouse serum.** The above results prompted us to search for beta cell trophic factors present in young blood that could stimulate adult beta cell proliferation. Using commercial antibody arrays that covered a total of 365 different proteins, we compared serum from p14 and 20wo mice. Applying a 2-fold cut-off, 17 proteins appeared to be more abundant in young than in adult serum (Supplementary Fig. 2). Some of the identified factors had been previously associated with beta cell growth, survival or function, including MCP-1[29], Igf-1[30,31], osteopontin[32,33], and osteoprogerin[34]. However, because all these proteins can be produced by islets[29,32,34,35], they could not be genuinely regarded as extrinsic factors.

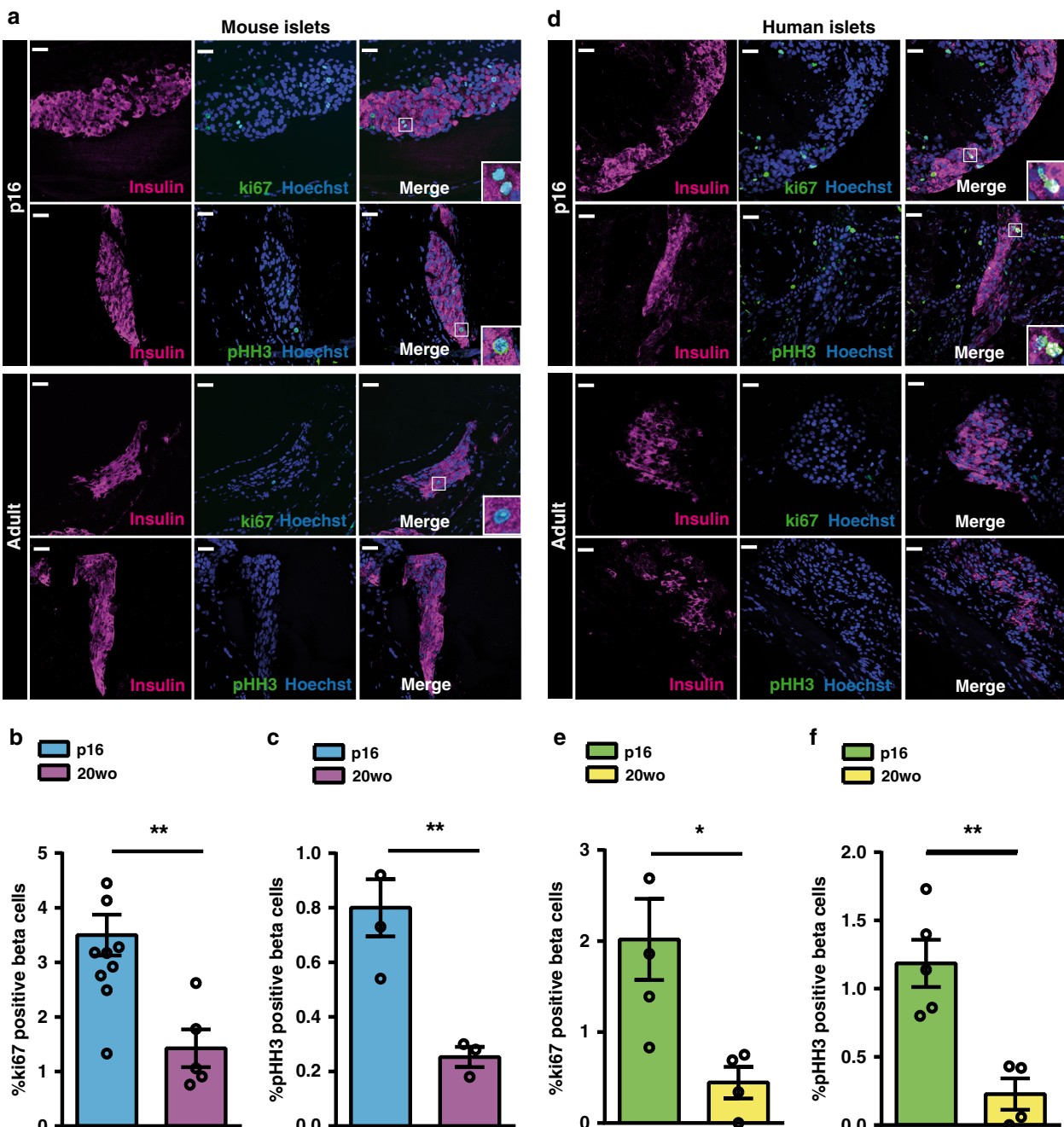

**Fig. 1 Adult beta cells exhibit enhanced replication when transplanted into pre-weaning mice. a–c** Beta cell replication of 20wo mouse islet grafts 12-days after implantation into the anterior chamber of the eye of p16 or 20wo C57BL6/J recipients. **a** Representative images of islet grafts co-immunostained for insulin (purple) /ki67 (green) or insulin (purple)/pHH3 (green). Nuclei are marked with Hoechst in blue. **b** Quantification of the percentage of beta (insulin+) cells that are ki67+ in islet grafts transplanted into p16 ($n = 11$, blue) and 20wo ($n = 5$, purple) mice. **c** Quantification of the percentage of beta (insulin+) cells that are pHH3+ in islet grafts transplanted into p16 ($n = 4$, blue) and 20wo ($n = 3$, purple) mice. **d–f** Beta cell replication of adult human islet grafts 12-days after implantation into the anterior chamber of the eye of p16 or 20wo NSG-SCID mouse recipients (islets from two donors). **d** Representative images of islet grafts co-immunostained for insulin (purple)/ki67 (green) or insulin (purple)/pHH3 (green). Nuclei are marked with Hoechst in blue. **e, f** Quantification of the percentage of beta (insulin+) cells that are ki67+ (**e**) or pHH3+ (**f**) in human islet grafts transplanted into p16 ($n = 5$, green) or 20wo ($n = 4$, yellow) mice. Data shown represent mean ± SEM for the indicated n. *$p < 0.05$; **$p < 0.01$, using two-tailed Student's $t$ test. Scale bars are 25 μm.

The CCN family protein CCN4/Wisp1 was also found enriched in p16 relative to 20wo serum (Fig. 2a). No information on the role of Wisp1 in pancreatic beta cells was available and hence we sought to interrogate the potential function of Wisp1 as a beta cell trophic factor. As an initial experiment we validated the array results with an ELISA specific for mouse Wisp1. We measured serum Wisp1 concentrations in mice of different ages (from p14 to 20wo) and confirmed an age-dependent decrease in circulating Wisp1 levels, which dropped nearly 6-fold between pre-weaning and adult stages (Fig. 2b). To corroborate these findings in humans, we measured circulating WISP1 with an ELISA specific for human WISP1 in blood from children aged 2–5 years and

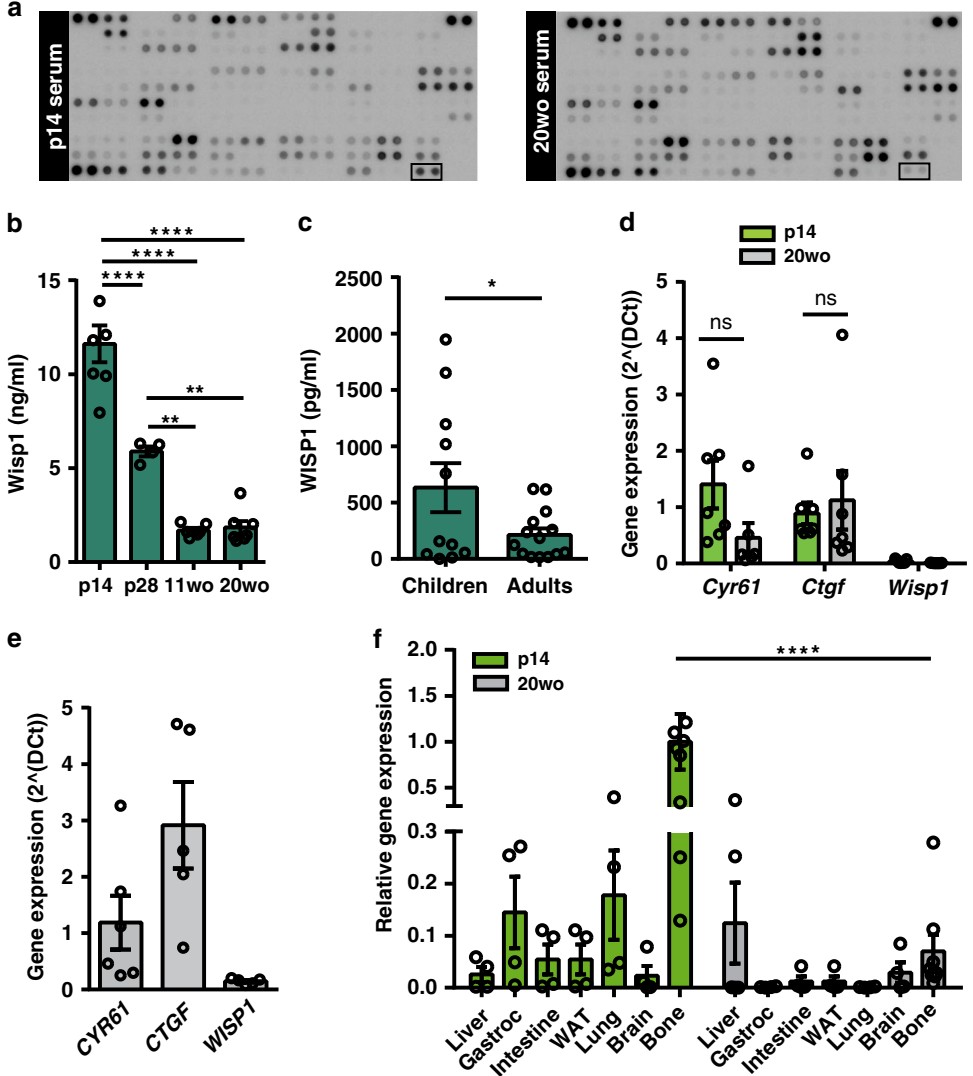

**Fig. 2 Identification of Wisp1 as a factor enriched in blood of pre-weaning mice. a** Representative images of antibody arrays incubated with p14 or 20wo mouse serum. Spots corresponding to Wisp1 are marked with an empty square box. **b** Serum levels of Wisp1 in p14 ($n = 7$), p28 ($n = 4$), 11wo ($n = 6$) and 20wo ($n = 7$) mice measured with ELISA. **c** Plasma levels of WISP1 in children ($n = 11$) and adult ($n = 14$) measured with ELISA. **d** Quantification by qPCR of the expression of the indicated CCN genes in p14 ($n = 7$, green) and 20wo mouse islets ($n = 6$ for *Cyr61*, $n = 7$ for other genes, gray). Values are expressed relative to *Tbp*. **e** Quantification by qPCR of the expression of the indicated CCN genes in adult human islets ($n = 6$ for *CYR61*, $n = 5$ for other genes). Values are expressed relative to *TBP*. **f** Quantification by qPCR of *Wisp1* gene expression in the indicated p14 (green) and 20wo (gray) mouse tissues. *Wisp1* gene expression is shown relative to levels in p14 bone, given the value of 1 ($n = 9$ for bone, $n = 4$ for all other tissues). WAT: white adipose tissue; Gastroc: gastrocnemius. All data shown represent mean ± SEM from the indicated n. Indicated comparisons were made using two-tailed Student's *t* test (**c**, **d**), one-way (**b**) and two-way (**f**) ANOVA. *$p < 0.05$; **$p < 0.01$; ****$p < 0.0001$; ns: not significant. In **f**, at p14, bone *Wisp1* gene expression was significantly higher than in all other tissues tested with $p < 0.01$–$0.001$.

adults aged 28–45 years. Remarkably, in concordance with our results in mice, WISP1 levels dropped 3-fold between child and adult stages (Fig. 2c).

Prior to investigating the role of Wisp1 in beta cells, we sought to ascertain the extrinsic nature of any potential effect of Wisp1 in islets. With this aim, we examined *Wisp1* gene expression in islets isolated from pre-weaning and adult mice. *Wisp1* transcript levels were negligible at either age thus discarding any potential role of Wisp1 as an intrinsic regulator (Fig. 2d). On the contrary, transcripts for the CCN genes *Cyr61* and *Ctgf* were readily detectable in mouse islets at both ages (Fig. 2d). Importantly, a similar expression pattern was observed in adult human islets (Fig. 2e).

To identify the tissue source of Wisp1 in suckling mice, we determined *Wisp1* mRNA levels in several p14 mouse tissues.

Among them, bone displayed the highest *Wisp1* gene expression levels followed by lung and skeletal muscle. *Wisp1* mRNA levels decayed in adult bone reaching values closer to the other surveyed tissues (Fig. 2f). Using immunohistochemistry, we found increased Wisp1 staining in young *versus* adult bone tissue (Supplementary Fig. 3). Corroborating this finding, calvaria bone cells isolated from young mice secreted more Wisp1 than those isolated from adult mice (Supplementary Fig. 3). Thus, these evidence indicate that the bone is the likely source of circulating Wisp1 in pre-weaning mice.

**Wisp1 contributes to proliferation of endogenous and transplanted beta cells in young mice.** To study the involvement of Wisp1 in beta cell proliferation during early postnatal life, we performed three experiments using mice with constitutive

deletion of the *Wisp1* gene (*Wisp1⁻/⁻*). First, we studied endogenous beta cell proliferation in fixed pancreatic tissue and found that beta cell staining of ki67 and pHH3 were both reduced in p14 *Wisp1⁻/⁻* mice as compared to their wild-type (*Wisp1⁺/⁺*) littermates (Fig. 3a–c). Circulating levels of Igf-1 and osteopontin were comparable whilst osteoprogeterin levels were modestly reduced (17%) in p14 *Wisp1⁻/⁻* as compared to their *Wisp1⁺/⁺* littermates (Supplementary Fig. 4a–c), indicating that the effects of *Wisp1* ablation on beta cell proliferation are unlikely secondary to alterations in these factors.

Second, we determined if administration of recombinant mouse Wisp1 (rmWisp1) to *Wisp1⁻/⁻* animals could enhance postnatal beta cell proliferation in vivo. To this aim, three intraperitoneal injections of rmWisp1 or saline were dispensed to *Wisp1⁻/⁻* pups for three consecutive days, from p9 to p11. At p12, *Wisp1⁻/⁻* pups that had received rmWisp1 presented proliferation rates that were 1.7–2 fold higher than those of littermate *Wisp1⁻/⁻* that had received saline (Fig. 3d–f). Note that proliferation rates in these latter group were higher than those reported for p14 *Wisp1⁻/⁻* pups (Fig. 3a, b), consistent with the rapid drop in beta cell proliferation that occurs during the first postnatal weeks[36,37]. No changes in Igf-1, osteoprogeterin or osteopontin were observed in mice treated with rmWisp1 as compared to saline controls (Supplementary Fig. 4d–f).

Third, to evaluate the contribution of Wisp1 to the replication of adult beta cells transplanted into young mice (Fig. 1), we

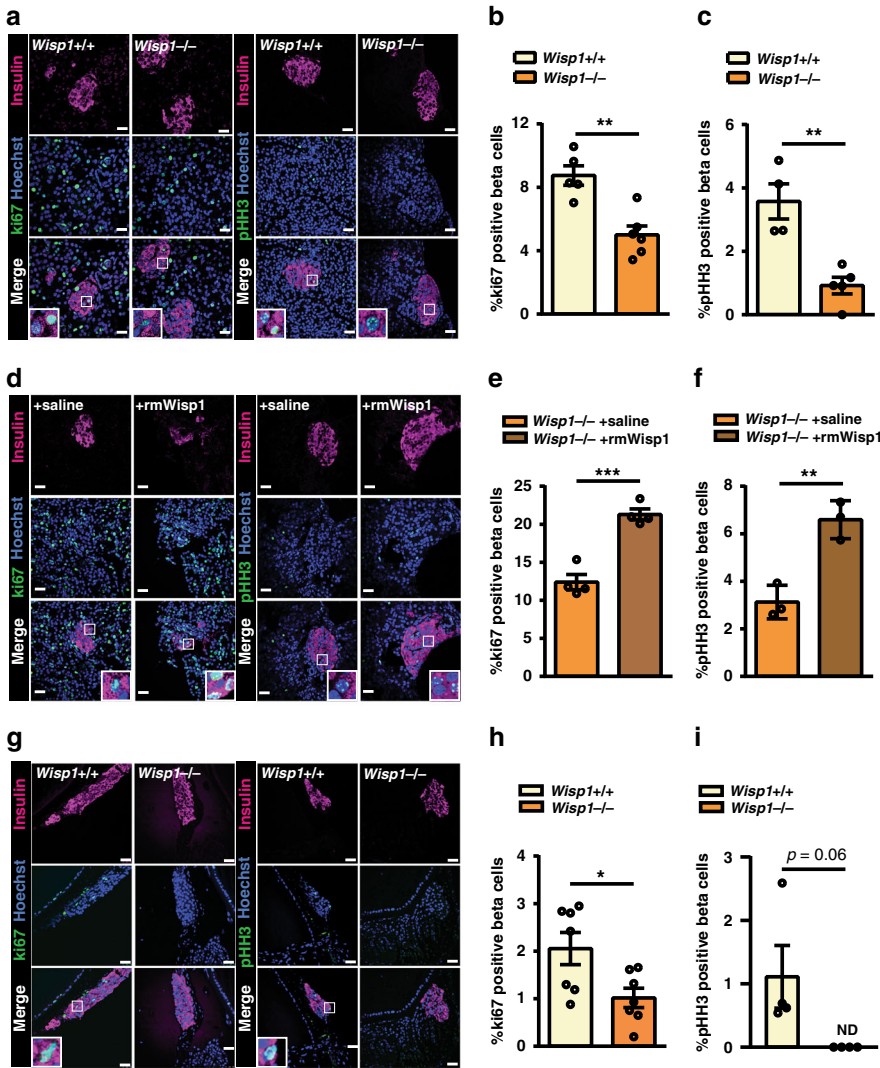

**Fig. 3 Wisp1 contributes to beta cell replication in young mice. a–c** Beta cell proliferation in fixed pancreases from p14 *Wisp1⁺/⁺* or *Wisp1⁻/⁻* mice. **a** Representative images of pancreases co-immunostained for insulin (purple)/ki67 (green) or insulin (purple) /pHH3 (green). Nuclei are marked with Hoechst in blue. **b** Quantification of the percentage of beta (insulin+) cells that are ki67+ in *Wisp1⁺/⁺* (n = 5, yellow) or *Wisp1⁻/⁻* (n = 6, orange) mice. **c** Quantification of the percentage of beta (insulin+) cells that are pHH3+ in *Wisp1⁺/⁺* (n = 4, yellow) or *Wisp1⁻/⁻* (n = 5, orange) mice. **d–f** Beta cell proliferation in fixed pancreases from p12 *Wisp1⁻/⁻* mice treated with saline or with rmWisp1 protein for three days (from p9 to p11). **d** Representative images of pancreases co-immunostained for insulin (purple)/ki67 (green) or insulin (purple)/pHH3 (green). Nuclei are marked with Hoechst in blue. **e** Quantification of the percentage of beta (insulin+) cells that are ki67+ in mice injected with rmWisp1 (n = 4, orange) o saline (n = 4, brown). **f** Quantification of the percentage of beta (insulin+) cells that are pHH3+ in mice injected with rmWisp1 (n = 3, orange) o saline (n = 3, brown). **g–i** Beta cell proliferation of 20wo mouse islet grafts transplanted into the anterior chamber of the eye of p16 *Wisp1⁺/⁺* or *Wisp1⁻/⁻* mouse recipients. **g** Representative images of islet grafts co-immunostained for insulin (purple)/ki67 (green) or insulin (purple)/pHH3 (green). Nuclei are marked with Hoechst in blue. **h** Quantification of the percentage of beta (insulin+) cells that are ki67+ in p16 *Wisp1⁺/⁺* (n = 7, yellow) or *Wisp1⁻/⁻* (n = 7, orange) mice. **i** Quantification of the percentage of beta (insulin+) cells that are pHH3+ in p16 *Wisp1⁺/⁺* (n = 4, yellow) or *Wisp1⁻/⁻* (n = 4, orange) mice. All data values represent mean ± SEM for the indicated n. *p < 0.05; **p < 0.01 using two-tailed Student's t test. Scale bars are 25 µm. ND: not detectable.

performed new transplants using p16 Wisp1$^{-/-}$ mice as recipients. We observed that beta cells of transplanted adult mouse islets proliferated less (50%) when implanted in Wisp1$^{-/-}$ relative to islets implanted in p16 Wisp1$^{+/+}$ littermates (Fig. 3g–i); and this was not due to differences in functional vascularization of the islet grafts between genotypes (Supplementary Movies 7–10).

Altogether, these results support the pro-proliferative role of Wisp1 in beta cells of pre-weaning mice.

**Wisp1 enhances beta cell proliferation and beta cell mass expansion in adult mice.** We next queried whether Wisp1 could promote proliferation of beta cells in adult mice. To overexpress this protein systemically, we administered adenoviral particles encoding the human isoform of WISP1 (Ad-WISP1) via tail vein injection to adult mice. As control, we injected a recombinant adenovirus encoding beta-galactosidase (Ad-betaGal). Human WISP1 transcripts were readily detected by qPCR in the liver at

days 7 and 14 post-administration of Ad-WISP1 (Fig. 4a). Hepatic mRNA levels of the mouse Wisp1 gene were not changed by transgenic expression of WISP1 (Fig. 4b). Using an ELISA specific for human WISP1, we confirmed the presence of the human protein in serum from mice injected with Ad-WISP1 but not in mice injected with Ad-betaGal (Fig. 4c). We then monitored the number of ki67-positive beta cells and observed that, 7 days after the adenoviral injection, mice receiving Ad-WISP1 presented higher beta cell proliferation as compared to mice receiving Ad-betaGal (Fig. 4d, e). This rise in beta cell replication resulted in a modest increase in beta cell mass 14 days post-injection of Ad-WISP1 (Fig. 4f), which did not have effects in either body weight or whole-body glucose tolerance (Supplementary Fig. 5).

We then investigated the potential of Wisp1 to promote beta cell proliferation in a pathological setting. To this aim, we administered Ad-WISP1 or Ad-betaGal to Stz-induced diabetic adult mice (Fig. 5a). Human WISP1 transcripts were detected in

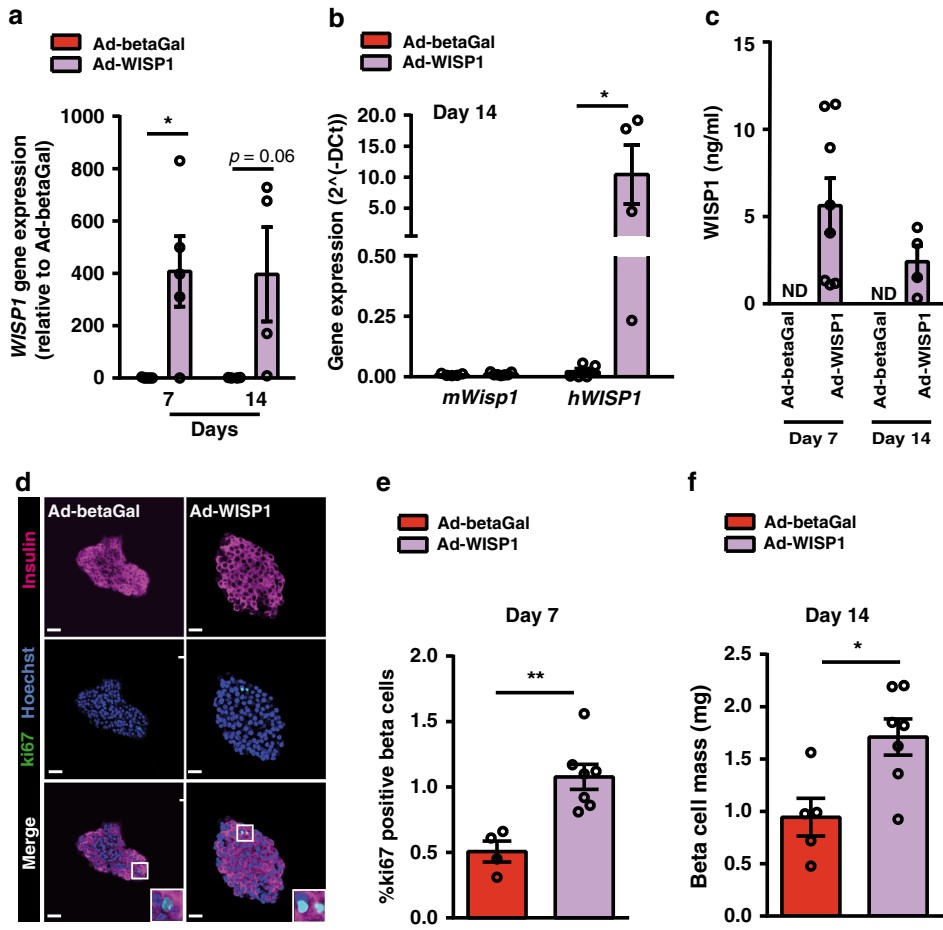

**Fig. 4 Adenovirus-mediated expression of Wisp1 enhances endogenous beta cell proliferation in adult mice.** Adenoviruses encoding human WISP1 (Ad-WISP1) or beta-galactosidase (Ad-betaGal) were injected via the tail vein into 12wo C57BL6/J mice. **a** Quantification by qPCR of human WISP1 transcripts in the livers of mice seven days and fourteen days ($n = 4$) post-injection. Levels are expressed relative to values in mice injected with Ad-betaGal, given the value of 1. **b** Quantification by qPCR of mouse Wisp1 mRNA ($n = 5$ for Ad-betaGal, red; $n = 7$ for Ad-WISP1, purple) and human WISP1 transcripts ($n = 5$ for Ad-betaGal, red; $n = 4$ for Ad-WISP1, purple) in the livers of mice fourteen days post-injection. Expression levels are expressed relative to Tbp. **c** Serum human WISP1 levels were measured by ELISA at days 7 and 14 post-injection of Ad-betaGal ($n = 7$) or Ad-WISP1 ($n = 8$ at day 7; $n = 4$ at day 14, purple). Human WISP1 was not detectable (ND) in serum from mice injected with Ad-betaGal. **d, e** Beta cell proliferation following injection of Ad-WISP1 and Ad-betaGal. **d** Representative images of in toto immunofluorescence staining against ki67 (green) and insulin (purple) in islets isolated at day 7 after injection of the indicated adenoviruses. Nuclei are labeled with Hoechst (blue). **e** Percentage of beta cells (insulin+) that are ki67+ at day 7 after injection of Ad-betaGal ($n = 4$, red) or Ad-WISP1 ($n = 7$, purple). **f** Beta cell mass at day 14 following injection of Ad-betaGal ($n = 5$, red) or Ad-WISP1 ($n = 7$, purple). All data shown represent mean ± SEM for the indicated n. *$p < 0.05$; **$p < 0.01$ using two-tailed Student's t test (**b, e, f**) or two-way ANOVA (**a, c**). Scale bars are 25 μm.

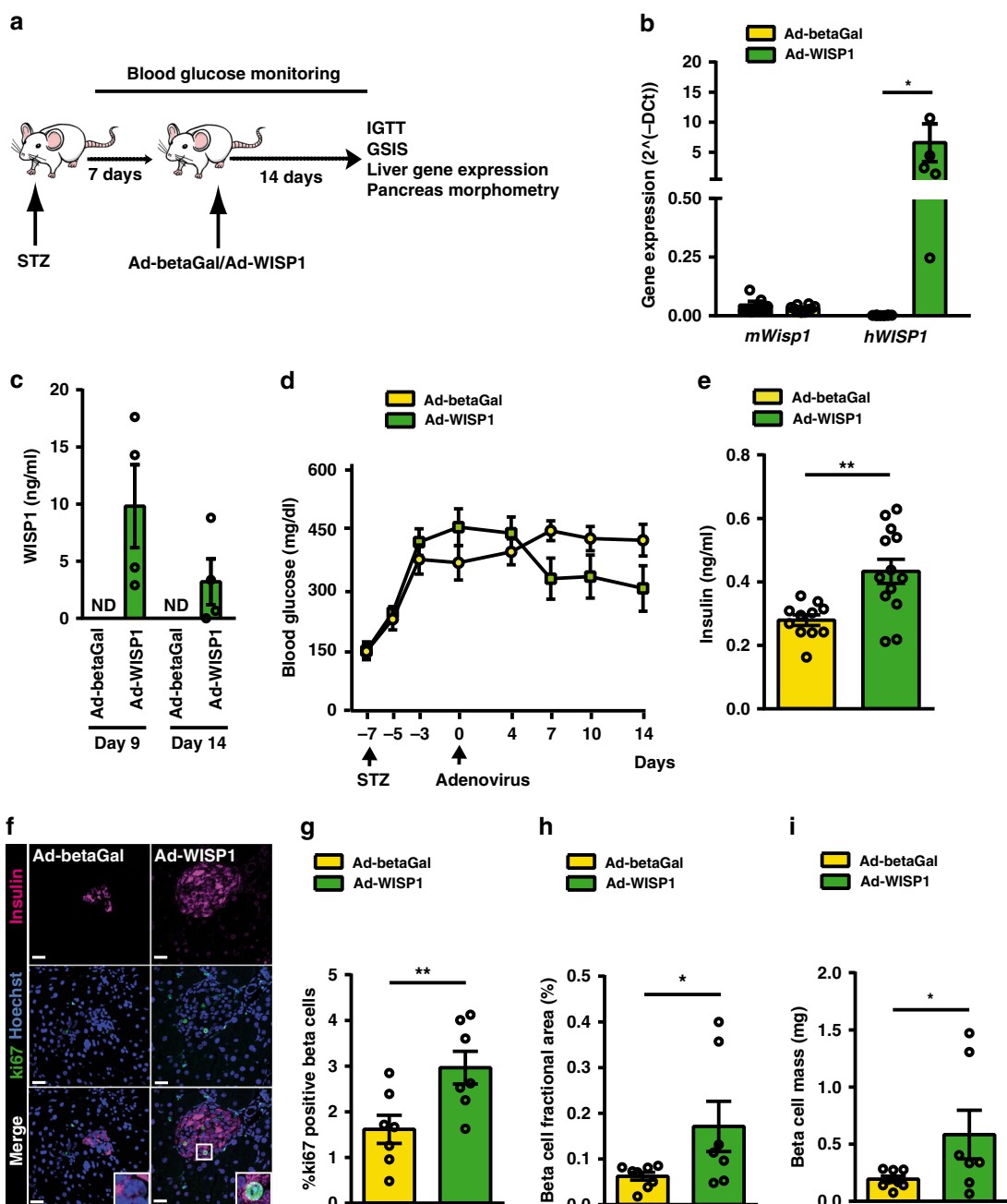

**Fig. 5 Adenovirus-mediated expression of Wisp1 promotes endogenous beta cell proliferation in adult diabetic mice. a** Schematic of experimental plan. **b** Quantification by qPCR of mouse *Wisp1* mRNA ($n = 6$ for Ad-betaGal, yellow; $n = 10$ for Ad-WISP1, green) and human *WISP1* transcripts ($n = 8$ for Ad-betaGal, yellow; $n = 6$ for Ad-WISP1, green) in the livers of mice fourteen days post-injection. Expression levels are expressed relative to *Tbp*. **c** Serum human WISP1 levels were measured by ELISA at days 9 and 14 post-injection with Ad-betaGal ($n = 5$) or Ad-WISP1 ($n = 4$, green). Human WISP1 was not detectable (ND) in mice injected with Ad-betaGal. **d** Blood glucose concentrations measured at the indicated days post-injection with Ad-betaGal ($n = 8$, yellow) or Ad-WISP1 ($n = 7$, green). **e** Serum insulin at day 14 following administration of Ad-betaGal ($n = 11$, yellow) or Ad-WISP1 ($n = 13$, green). **f, g** Beta cell proliferation following injection of Ad-WISP1 and Ad-betaGal. **f** Representative images of immunofluorescence staining against ki67 (green) and insulin (purple) in fixed pancreases at day 14 after injection of the indicated adenoviruses. Nuclei are labeled with Hoechst (blue). **g** Percentage of beta cells (insulin+) that are ki67+ at day 14 days after injection of the indicated adenoviruses ($n = 7$; Ad-betaGal in yellow, Ad-WISP1 in green). **h** Beta cell fractional area (insulin+ area relative to total pancreatic area) and **i** total beta cell mass at day 14 after injection of Ad-betaGal ($n = 8$, yellow) or Ad-WISP1 ($n = 7$, green). All data shown represent mean ± SEM for the indicated n. *$p < 0.05$; **$p < 0.01$ using two-tailed Student's t test (**b**, **e**), one-tailed Student's t test (**g–i**) and two-way ANOVA (**d**). Scale bars are 25 μm.

the liver 14 days post-injection with Ad-WISP1 without accompanying changes in endogenous *Wisp1* transcripts (Fig. 5b). The presence of human WISP1 in serum of Ad-WISP1 injected mice was confirmed at days 9 and 14 post-viral injection (Fig. 5c). Systemic overexpression of WISP1 did not normalize Stz-induced

hyperglycemia during the follow-up period, even though there was a trend to lower glucose levels (Fig. 5d). However, at day 14 after injection with Ad-WISP1, mice exhibited a significant increase in ad lib fed plasma insulin levels as compared to animals injected with Ad-betaGal (Fig. 5e). This change was not

associated with amelioration of whole-body glucose tolerance or glucose-induced insulin secretion in Ad-WISP1-injected mice (Supplementary Fig. 6). Yet, detailed morphometric assessment of the pancreatic beta cell compartment revealed a nearly two-fold increase in proliferating beta cells in animals treated with Ad-WISP1 relative to controls (Fig. 5f, g). In line with enhanced proliferation, we found increased insulin-positive area and total beta cell mass (Fig. 5h, i).

Collectively, these experiments confirm that circulating WISP1 can increase the proliferation of pancreatic beta cells in situ in adult animals, further supporting the potential of this protein as a beta cell trophic factor.

**Wisp1 stimulates mouse and human beta cell proliferation in vitro.** All of the preceding experiments were performed using in vivo models. Here, we sought to investigate if Wisp1 could induce adult beta cell proliferation directly, in the absence of other systemic factors. First, we treated adult mouse islets ex vivo for 2 days with different concentrations of recombinant mouse Wisp1 (rmWisp1) protein and determined the percentage of ki67-positive beta cells. As shown in Fig. 6a, rmWisp1 induced a a dose-dependent rise in beta cell proliferation, with a maximum 2.4-fold increase observed at 500 ng/ml rmWisp1 relative to control values. The effect of rmWisp1 was compared to that of the small molecule harmine, a DYRK1A inhibitor and c-myc activator[38]. Correlating with previous studies[38,39], harmine promoted a dramatic increase in mouse beta cell proliferation, reaching values 4-fold higher than with rmWisp1 (Fig. 6a, b).

Recombinant factor-based assays typically require supraphysiological amounts of protein to maintain effective concentrations; hence we performed a second experiment to validate the mitogenic effect of physiological quantities of Wisp1. To this aim, we treated NIH3T3 cells with Ad-WISP1 or Ad-betaGal and then co-cultured them with adult mouse islets for 24 h or 48 h. In this experiment the achieved concentration of human WISP1 in the culture media (3–6 ng/ml) was in the same order of magnitude than endogenous Wisp1 in mouse serum (Supplementary Fig. 7). Consistent with experiments with rmWisp1, beta cells in mouse islets exposed to NIH3T3/Ad-WISP1 presented enhanced ki67 labeling as compared to beta cells in islets co-cultured with NIH3T3/Ad-betaGal (Fig. 6c, d). In agreement with this observation, islets exposed to NIH3T3/Ad-WISP1 had increased expression of known Wisp1 gene targets such as *Axin2* or *Ndrg1/3* and of the cell cycle genes *Foxm1* and *Plk1* (Supplementary Fig. 7).

Human beta cells have an apparent lower replicative potential than mouse beta cells, and different factors described in the literature as inducers of beta cell replication in rodents have been reported to exert modest or no effect in human beta cells. Thus, we interrogated the proliferative effects of recombinant human WISP1 (rhWISP1) in human islets. A two-day treatment with rhWISP1 increased dose-dependently the percentage of doubly positive ki67/insulin cells (Fig. 6e, f). Remarkably, the effectiveness of rhWISP1 was comparable to that of harmine in human islets (Fig. 6f), reaching proliferation values of ~2% similar to those described for human beta cells during the first year of life[40]. The reduced effectiveness of harmine in human as compared to mouse islets has been previously reported[38,39].

**Wisp1 acts via AKT to stimulate mouse and human beta cell proliferation in vitro.** Akt/PKB has been postulated as the intracellular mediator of Wisp1 actions in several cellular contexts[41,42]. To address the involvement of Akt in Wisp1-induced beta cell proliferation, we first determined levels of *Ser473* phosphorylation of this kinase (active form) in mouse

islets using immunoblot analysis. Incubation with rmWisp1 significantly increased phospho-Akt levels (Fig. 7a), thus corroborating that this kinase is a downstream target of Wisp1 in islets. Furthermore, chemical inhibition of Akt impaired Wisp1-induced beta cell proliferation (Fig. 7b, c), proving the involvement of Akt in the pro-proliferative actions of Wisp1.

Finally, we queried whether Akt was also required for the mitogenic effect of Wisp1 in human islets. Like their mouse counterparts, incubation with rhWISP1 augmented AKT phosphorylation in human islets and chemical inhibition of this kinase blocked WISP1-triggered increase in the number of ki67+ human beta cells (Fig. 7d, e).

Together, these results support that Akt mediates the pro-proliferative effects of Wisp1 in mouse and human beta cells.

## Discussion

This study supports the involvement of systemic factors in the decline of beta cell proliferation during normal organism growth. Our present results parallel prior parabiosis experiments showing that circulating factors in young adult mice (1-month old) increased beta cell proliferation rates in older mice (8 months old)[7]. Here, by grafting mouse and human islets in the eye, we demonstrate the existence of systemic factors in pre-weaning mice, a time when beta cell replication is highly active, that can stimulate adult beta cell proliferation. These observations add to an emerging body of evidence showing the regenerative capacity of young blood in other organs such as skeletal muscle[43], heart[44] and brain[45]. In our search of serum proteins contributing to high beta cell proliferation rates in suckling pups, we identified Wisp1 as a factor whose abundance in circulation drops after weaning in mice. Notably, we found that circulating WISP1 levels were also higher in children than in adults, revealing an analogous age-dependent decline of this factor in both species.

We found the bone as the tissue exhibiting highest *Wisp1* gene expression levels in suckling mice and confirmed higher production of Wisp1 by young than adult bone. These observations are in agreement with the reported involvement of Wisp1 in the regulation of skeletal remodeling and bone structure[22] and the profound changes in skeletal metabolism that occurs in suckling pups, as bone formation takes place mostly during embryonic development and postnatal growth[46]. In fact, the skeleton is no longer recognized only as a mechanical scaffold, but also as an endocrine organ which regulates glucose metabolism and energy expenditure[47,48]. It is also known to secrete humoral factors that affect beta cell mass, such as osteopontin[32], osteoprotegerin[34], and osteocalcin[49]. Remarkably, we found that both osteopontin and osteoprotegerin were increased in serum of pre-weaning mice, reinforcing the notion that bone produces blood-borne factors that participate in early postnatal beta cell mass expansion.

It will be interesting to determine whether Wisp1 is involved in known conditions of enhanced adult beta cell proliferation, such as insulin resistance and obesity. Intriguingly, Wisp1 has been recently described as an adipokine whose levels negatively correlate with insulin sensitivity[25], although no differences in circulating Wisp1 levels have been reported between normal, impaired glucose tolerant and diabetic human subjects[50]. We have examined serum Wisp1 levels in mice fed a high fat diet (HFD) and found a rather modest increase (~2-fold) after 10 day HFD feeding, but no differences after 60 or 120 day HFD feeding (Supplementary Fig. 8), indicating that, if any, the role of Wisp1 might be limited to the reported initial activation of beta cell proliferation in response to HFD[51].

One central result of this study is that Wisp1 promotes proliferation of human beta cells, which are considered refractory to the mitogenic actions of many factors that exert robust pro-

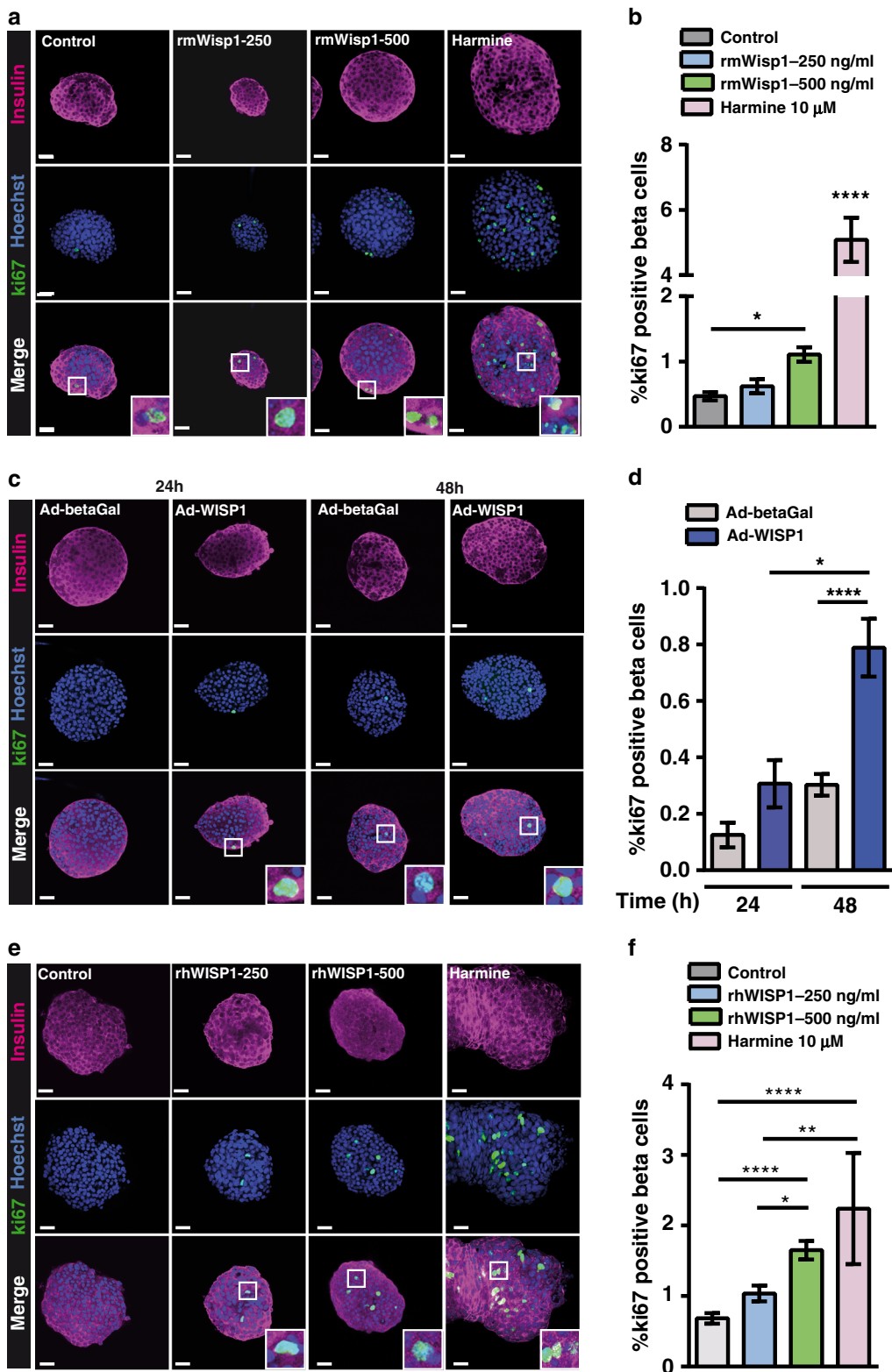

proliferative effects in mouse beta cells. Nonetheless, from a therapeutic standpoint, the existing studies that report mitogenic effects of Wisp1 in various cell types[20,41,52–54], raise concerns about its specificity and the risk that systemic administration of this factor might have undesirable effects in extra-pancreatic tissues. Yet, it has been described that Wisp1 does not signal through a unique specific receptor but it rather acts by modulating other growth factor signaling pathways[55], making Wisp1

actions dependent not only on intrinsic cues of its target cell but also on the specific extracellular context where it acts. Thus, combination of Wisp1 with other factors might be the key to provide enhanced specificity towards beta cells. For instance, in agreement with reports in other cellular contexts[41,56], here we show that Wisp1 activates Akt in mouse and human islets. That Akt activation induces beta cell proliferation has been previously established[57,58]. Yet, this effect is often connected with its

**Fig. 6 Wisp1 promotes mouse beta cell replication in vitro. a, b** Beta cell proliferation in mouse islets incubated with Wisp1 recombinant mouse protein (rmWisp1). **a** Representative images of in toto immunofluorescence showing ki67 (green) and insulin (purple) staining in mouse islets cultured for 48 h with increasing amounts of rmWisp1 protein or harmine. Nuclei are marked with Hoechst in blue. **b** Percentage of beta cells (insulin+) that are ki67+ in cultured islets under the indicated conditions (control, $n = 49$ islets, in gray; rmWisp1-250, $n = 19$ islets, in blue; rmWisp1-500, $n = 45$ islets, in green; harmine, $n = 15$ islets, in pink; from four independent experiments except for harmine that are from two independent experiments). **c, d** Beta cell proliferation in mouse islets co-cultured with NIH3T3 cells expressing WISP1. **c** Representative images of in toto immunofluorescence showing ki67 (green) and insulin (purple) staining in mouse islets co-cultured for 24 or 48 h with NIH3T3 cells infected with the indicated adenoviruses. Nuclei are marked with Hoechst in blue. **d** Percentage of beta cells (insulin+) that are ki67+ in mouse islets cultured under the indicated conditions for 24 h (control, in gray: $n = 25$ islets; 3T3/WISP1, in blue: $n = 21$ islets, from three independent experiments) and for 48 h (control, in gray: $n = 100$ islets; 3T3/WISP1, in blue: $n = 88$ islets, from three independent experiments). **e, f** Beta cell proliferation in human islets incubated with recombinant human WISP1 protein (rhWISP1). **e** Representative images of in toto immunofluorescence showing ki67 (green) and insulin (purple) staining in human islets incubated for 48 h with increasing amounts of rhWISP1 protein or with harmine. Nuclei are marked with Hoechst in blue. **f** Percentage of beta cells (insulin+) that are ki67+ in human islets cultured under the indicated conditions (control, in gray: $n = 79$ islets; rhWISP1-250, in blue: $n = 60$ islets; rhWISP1-500, in green: $n = 98$ islets; harmine, in pink: $n = 17$ islets; from four donors). All data shown represent mean ± SEM for the indicated n. Comparisons were made using one-way ANOVA. *$p < 0.05$; **$p < 0.01$; ****$p < 0.0001$. Scale bars are 25 μm.

function as a central mediator of the Insulin/IGF signaling pathway, which is an important regulator of beta cell growth[59–61]. Therefore, it is tempting to speculate that Wisp1 might act synergistically with ligands of the Insulin/IGF pathway to activate beta cell proliferation via modulation of Akt activity. Future studies should be aimed at addressing the beta cell trophic effects of this and other possible factor combinations.

In conclusion, our study identifies Wisp1 as a circulating protein that is abundant in young blood and induces proliferation of adult beta cells, thus revealing Wisp1 as an agent with potential therapeutic use to expand beta cell mass in diabetes. This work lays the background for further studies aimed at identifying circulating factors present in early postnatal life that can be used as therapeutic agents to expand beta cell mass in adulthood.

## Methods

**Animals**. C57BL6/J, immunocompromised NSG-SCID (Jackson Laboratories) and Wisp1 knockout mice (Wisp1$^{-/-}$)[22] were bred at the barrier animal facility of the University of Barcelona. Wisp1$^{-/-}$ mice were genotyped with primers provided in Supplementary Table 1. Animals were maintained in a 12-h light/12-h dark cycle in temperature and humidity-controlled environment with free access to water and standard laboratory chow. All adult mice used in the experiments were male. Pre-weaning mice were both male and female.

Diabetes was induced in 4–6 h fasted 12–16 week-old C57BL6/J male mice by a single intraperitoneal injection of 125 mg/Kg streptozotocin (Stz, Sigma-Aldrich, St Louis, MO, USA) diluted in 0.9% NaCl with 100 mmol/l sodium citrate (pH 4.5). Mice that had plasma blood glucose concentrations above 300 mg/dl for three consecutive days were considered for the experiment.

**Antibody arrays**. Serum samples were individually obtained by centrifugation of the whole blood at 8000 × $g$ for 15 min twice. Three different p14 and 20wo serum batches (each a pool from 6 animals) were analyzed using the RayBio Mouse Cytokine Antibody Array Kit (Raybiotech, Inc. Georgia, USA) and Proteome Profiler$^{TM}$ Array/Mouse XL Cytokine Array Kit (R&D Systems, Minneapolis, USA) following manufacturer's instructions. Membranes were scanned using a LAS4000 Lumi-Imager (Fuji Photo Film, Valhalla, NY) and analyzed using ImageJ/Fiji software. The results were then normalized using internal controls, and the relative protein abundance in p14 compared to adult mouse serum was represented.

**Islet isolation and culture**. Donor mouse islets were obtained from C57BL6/J male mice with ages ranging from 12 to 20 weeks by collagenase digestion and Histopaque gradient purification as previously described[62]. Isolated islets were allowed an overnight recovery in RPMI-1640 media with 11 mM glucose, 10% FBS and antibiotics before performing experiments.

Human islets were obtained from 8 cadaveric donors (males and females) with an average age of 54.1 ± 3.7 years and BMI of 24.7 ± 1.4 kg/m$^2$. Isolated islets were prepared by collagenase digestion followed by density gradient purification at the Laboratory of Cell Therapy for Diabetes (Hospital Saint-Eloi, Montpellier, France), as previously described[63]. After reception, human islets were maintained in culture for 1–3 days in RPMI-1640 with 5.5 mM glucose, 10% fetal bovine serum (FBS) and antibiotics, before performing the experiments.

**Transplant into the anterior chamber of the eye**. Groups of 100-150 mouse or human islets were introduced and allowed to deposit at the bottom of a polythene cannula (Fine Bore Polythene Tubing, Smiths Medical International). Recipient mice (C57BL6/J, Wisp1$^{+/+}$, Wisp1$^{-/-}$ or NSG-SCID) were anesthetized with ketamine-xylacine (100 mg/Kg body weight Ketamine+10 mg/ Kg body weight Xylacine), and islets were introduced in the anterior chamber of the eye under a stereomicroscope, using a Hamilton syringe. Viscofresh (10 mg/ml sodium carmellose) was used to relieve eye discomfort when the mice recovered from the anesthetics. Twelve days after implantation, mice were euthanized and eyes were removed and processed for subsequent immunofluorescence analysis.

Functional graft vascularization was assessed as described[64]. In brief, mouse islets were labeled with 10 μM CFDA SE (Vybrant® CFDA SE Cell Tracer Kit, Invitrogen$^{TM}$, Molecular Probes®) for 15 min at 37 °C. Mice were anesthetized with Ketamine (100 mg/kg)-Xylazine (10 mg/kg), and 80 mg/Kg body weight of rodamine dextrane (Rhodamine B Isothiocyanate-Dextran, Sigma) was administered intraocularly. Live observation of blood vessels in the anterior chamber of the eye was carried out under a Leica TCS SP5 resonant scan spectral confocal microscope (Leica Microsystems Heidelberg GmbH) equipped with an incubation system with temperature control, a HCX IR APO L 25× water immersion objective (Numerical Aperture 0.95), resonant scanner at 8000 lines/s and blue diode (405 nm), Argon (488 nm) and diode pumped solid state (561 nm). Fluorescent dyes CFDA SE and Dextran-Rhodamine were acquired simultaneously using 488 and laser lines, AOBS (Acoustic Optical Beam Splitter) as beam splitter and emission detection ranges 500–550 nm and 571–625 nm, respectively and the confocal pinhole set at 4 Airy units. Z sectioning was performed every 300 nm to reconstruct whole islet.

**In vivo adenoviral treatment**. The recombinant adenovirus encoding full-length human WISP1 (Ad-WISP1) and the control adenovirus encoding beta galactosidase (Ad-betaGAL) were described elsewhere[65,66]. Adenoviruses were purified and titrated by the Viral Vector Production Unit of the University Autonomous of Barcelona. Recombinant adenoviruses (2.45xE9 pfu) were injected into the tail vein of both control and diabetic C57BL6/J mice. Blood glucose was regularly monitored in diabetic mice after viral injection. At the time of sacrifice, pancreases were harvested and processed for immunofluorescence staining.

**In vivo assessment of glucose homeostasis**. The Intraperitoneal Glucose Intolerance Test (ipGTT) was performed after 5–6 h of food deprivation by administration of an intraperitoneal injection of D-glucose (2 g/Kg body weight). Glucose in tail vein blood was measured at the indicated time points after injection using a clinical glucometer and Accu-Check test strips (Roche Diagnostics, Switzerland). Plasma samples were obtained by blood centrifugation and kept at −80 °C for subsequent insulin determination by ELISA.

**Gene expression analysis**. Total RNA was isolated from mouse tissues using TRI Reagent® (Sigma-Aldrich, Missouri, USA) and from isolated islets using the NucleoSpin XS RNA kit (Mackerey-Nagel, Düren, Germany). First-strand cDNA was prepared using the Superscript III RT kit and random hexamer primers (Invitrogen, Carlsbad, CA, USA). Reverse transcription reaction was carried for 90 min at 50 °C and an additional 10 min at 55 °C. Real time quantitative PCR (qPCR) was performed on an ABI Prism 7900 sequence detection system using GoTaq® qPCR Master Mix (Promega Biotech Ibérica, Alcobendas, Madrid, Spain). Expression relative to the housekeeping gene Tbp was calculated using the deltaCt method. Primer sequences are provided in Supplementary Table 1.

**Immunofluorescence staining and morphometric analysis**. Pancreases and eyes were harvested, fixed overnight in 10% formalin and 2% paraformaldehyde,

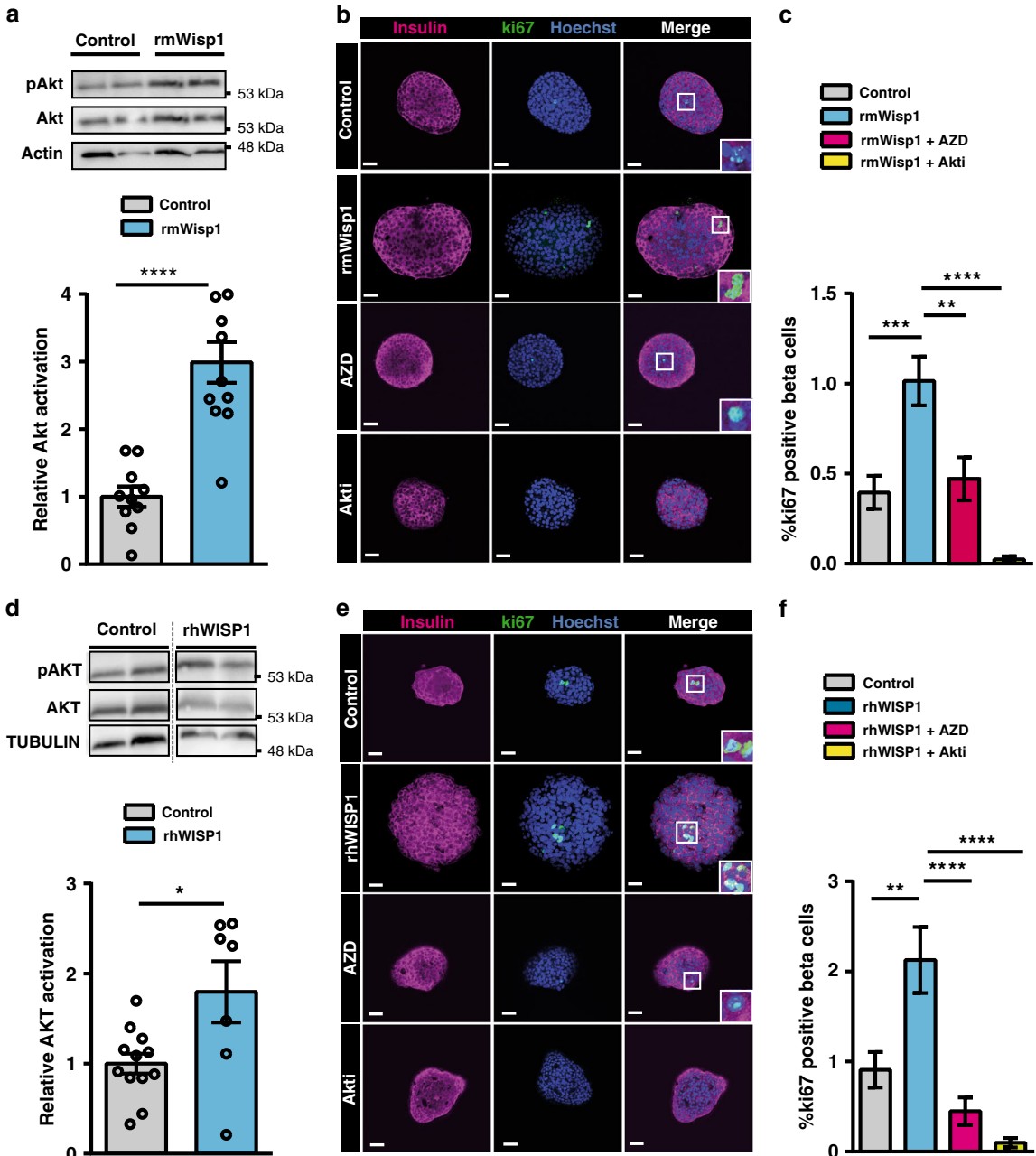

**Fig. 7 Wisp1 activates Akt in mouse and human pancreatic islets. a** Determination of Akt activation (*Ser473* phosphorylation) by immunoblot analysis in mouse islets incubated with recombinant mouse Wisp1 protein (rmWisp1) at 500 ng/ml for 30 min. Top: representative immunoblot image. Molecular weight markers are shown on the right. Bottom, quantification of Akt activation, expressed relative to control islets (no rmWisp1), given the value of 1 (control, in gray: $n = 10$; rmWisp1, in blue: $n = 11$, from five independent experiments). **b, c** Beta cell proliferation in mouse islets incubated with rmWisp1 protein and Akt inhibitors. **b** Representative immunofluorescence images showing ki67 staining in green and insulin in purple. Nuclei are marked with Hoechst (blue). **c** Percentage of beta cells (insulin+) that are ki67+ in islets incubated with rmWisp1protein at 500 ng/ml for 48 h alone ($n = 41$ islets, blue) or with the Akt inhibitors AZD5363 ($n = 21$ islets, pink) or Akti ($n = 25$ islets, yellow), or left untreated (control, gray: $n = 39$ islets) from three different isolation experiments. **d** Determination of AKT activation (*Ser473* phosphorylation) by immunoblot analysis in human islets incubated with recombinant human WISP1 protein (rhWISP1) at 500 ng/ml for 15 min. Top: representative immunoblot image. Molecular weight markers are shown on the right. Bottom, quantification of AKT activation, expressed relative to control islets (no WISP1), given the value of 1 (control in gray: $n = 12$; rhWISP1 in blue: $n = 7$, from 3 donors) **e, f** Beta cell proliferation in human islets incubated with Wisp1 protein and Akt inhibitors. **e** Representative immunofluorescence images showing ki67 staining in green and insulin in red. Nuclei are marked with Hoechst (blue). **f** Percentage of beta cells (ins+) that are ki67+ in human islets incubated with rhWISP1 protein at 500 ng/ml for 48 h alone ($n = 21$ islets, blue) or with the Akt inhibitors AZD5363 ($n = 31$ islets, pink) or Akti ($n = 15$ islets, yellow), or left untreated (control, gray: $n = 20$ islets) from 3 donors. All data shown represent mean ± SEM for the indicated $n$. Comparisons were made using two-tailed Student's $t$ test (**a, d**) or one-way ANOVA (**c, f**). *$p < 0.05$; ***$p < 0.001$; ****$p < 0.0001$. Scale bars are 25 μm.

respectively, and subsequently washed, dehydrated and embedded in paraffin wax. For immunofluorescence in paraffin sections, a standard immunodetection protocol was followed as described elsewhere[67]. Briefly, tissues were rehydrated and, when required, subject to heat-mediated antigen retrieval in citrate buffer. After a blocking step in 5% donkey or goat serum, and permeabilization using 1% (v/v) Triton X-100, tissue sections were incubated overnight with primary antibodies and then for 1 h with fluorescent-labeled secondary antibodies (Supplementary Table 2). For beta cell morphometric analysis, at least six non-consecutive (<150 μm apart) 4 μm thick sections were analyzed per pancreas. Beta cell fractional area was quantified as insulin-positive area relative to total pancreatic area. Beta cell mass was calculated as beta cell fractional area multiplied by pancreas weight. Beta cell proliferation was expressed as the percentage of cells doubly stained for insulin/ki67 or insulin/phosho-histone H3 (pHH3) relative to total number of insulin-positive cells, and at least 1500–3000 beta cells per animal were counted. Vascularization in transplanted islet grafts was calculated as the percentage of CD31+ area relative to insulin+ area.

For in toto immunofluorescence, whole islets were fixed in 4% paraformaldehyde for 30 min, washed with PBS, permeabilized in PBS-0.5% Triton X-100 and blocked in PBS-10% FBS-0.5% Triton X-100. Subsequently, islets were incubated overnight with primary antibodies and then for 2 h with fluorescent-labeled secondary antibodies (Supplementary Table 2). Nuclei were stained with Hoechst 33258 (Sigma). Fluorescent images were captured using a Leica TCS SPE confocal microscope, collected with Leica LAS AF v2.7.3.9723 and subsequently analyzed using Image J v1.50d software (Wayne Rayband, National Institutes of Health, http://rsb.info.nih.gov/ij/). For quantification of beta cell proliferation, the number of cells doubly positive for insulin and ki67 or pHH3 was analyzed in Z stacks 10 μm apart in order to avoid counting the same cell twice.

**NIH3T3 and mouse islet co-culture**. NIH-3T3 cells were grown in DMEM-4.5 g/L glucose (Sigma-Aldrich, St Louis, MO, USA) plus 10% Calf Serum and antibiotics. For adenoviral infection, NIH-3T3 cells were seeded onto 12-well plates (2.5xE4 cells/well) and incubated one day later with Ad-WISP1 or Ad-betaGal at a multiplicity of infection (moi) of 25 for 6 h. 12–16 h after virus removal, cells were co-cultured with mouse islets isolated the previous day. Co-cultures were performed in transwell culture inserts consisting of 0.4 μm translucent pet membrane (BDFalcon) using RPMI-1640 plus 8 mM glucose and 5% FBS. Following 24 and 48 h of co-culture, medium was collected for ELISA and islets were recovered for total RNA extraction and for in toto immunofluorescence staining to quantify proliferation.

**Recombinant Wisp1 protein**. Recombinant mouse Wisp1 and human WISP1 proteins were purchased from R&D Systems. Isolated mouse or human islets were cultured in RPMI-1640 plus 8 mM glucose and 5% FBS with or without the indicated recombinant proteins alone or with the Akt inhibitors AZD5363 (5 μM, Selleckchem) and Akti-1/2 (10 μM, Abcam) during the times indicated. Islets were then recovered for subsequent in toto immunofluorescence staining and immunoblot analysis.

For in vivo Wisp1 treatment, mouse recombinant mouse Wisp1 protein was administered daily by intraperitoneal injection at 1 mg/Kg body weight per day to Wisp1$^{-/-}$ pups from p9 to p11. At p12, one day after last injection, animals were anesthetized and blood samples and pancreases were stored for subsequent analysis of circulating recombinant Wisp1 protein, and immunohistological analysis of the pancreas of control and treated mice, respectively.

**Immunoblot analysis**. Islets were lysed in triple detergent lysis buffer (Tris-HCl 50 mM, NaCl 150 mM, 0.1% (w/v) SDS, 1% (v/v) NP40 and 0.5% (w/v) sodium deoxycholate). After 3 consecutive cycles of dry ice and 37 °C, cell debris was pelleted and supernatants were prepared for SDS-PAGE electrophoresis on 8% Tris-tricine homemade gels. Proteins were then transferred to a Polyscreen PVDF membrane (Perkin Elmer, Waltham, MA, USA) and incubated overnight at 4 °C with the antibodies indicated in Supplementary Table 2. Immunoblots were developed with horseradish peroxidase-conjugated secondary antibodies (1:5000, GE Healthcare Bio-Sciences Corp. Piscataway, NJ, USA) and visualized with ECL Reagent (Pierce Biotechnology, Rockford, IL, USA) using a LAS4000 Lumi-Imager (Fuji Photo Film, Valhalla, NY). Protein spots were quantitated with Image J software. Due to the little amount of protein obtained from islets, protein extracts were not quantified but instead, the same number of islets was loaded in each lane. Representative images of the quantification results were selected. Uncropped blots are supplied in the Source Data file.

**ELISAS**. Mouse Wisp1 levels were determined using a Mouse/Rat Wisp-1/CCN4 Immunoassay (R&D Systems) that detects both natural and full-length recombinant Wisp1 protein. ELISAS to determine serum levels of mouse osteopontin, mouse osteoprogeterin and mouse Igf-1 were purchased from R&D Systems (Abingdon, UK). Mouse insulin was analyzed using an Ultra Sensitive Mouse Insulin ELISA Kit (Crystal Chem, IL, USA). Human WISP1 levels in sera from mice injected with Ad-WISP1 and in archived human plasma samples were determined using a Human WISP1/CCN4 PicoKine™ ELISA Kit (Boster Biological Technology, California, USA). Specific antibodies for this ELISA were

produced using T23-N367 of human WISP1 as immunogen (full-length WISP1 is 367 aminoacids). A total of 11 samples originated from clinically healthy children, aged 2–5 years, and were obtained from the Biobank of the Hospital Infantil Sant Joan de Deu. A total of 14 samples originated from clinically healthy adult men, aged 28–45 years, and were obtained from the Hospital Clinic-IDIBAPS Biobank. Informed consent was obtained for all donors. All assays were performed following the manufacturer´s instructions.

**Statistical information**. Data were analyzed using GraphPad Prism version 6.00 for Windows (www.graphpad.com) and Microsoft Office Excel 2007, and expressed as the mean ± standard error of the mean (SEM). The appropriate statistical test was determined based on the number of comparisons being done. Student's $t$-tests (two-tailed unless otherwise stated) were used for comparison of two groups. One-way ANOVA was used for analysis of three or more experimental groups. Two-way ANOVA was used for analysis of adenoviral WISP1 gene overexpression, mouse serum Wisp1 concentrations and ipGTT tests in in vivo overexpression experiments. When finding significant results in ANOVA, recommended post-hoc tests (Tukey or Sidak) were applied to determine significant differences among multiple comparisons. $P < 0.05$ was deemed significant. $P$ values greater than 0.05 and lower than 0.1 were considered trends and exact values are shown in the figures.

**Ethics statement/study approval**. All animal procedures were approved by the Animal Ethics/Research Committee of the University of Barcelona. Principles of laboratory animal care were followed (European and local government guidelines). Experiments involving human islets were performed in agreement with the local ethic committee (CHU, Montpellier) and the institutional ethical committee of the French Agence de la Biomédecine (DC Nos. 2014-2473 and 2016-2716). Informed consent was obtained for all human islet donors. Human blood samples were obtained from the biobanks of the Hospital Infantil Sant Joan de Deu and Hospital Clinic-IDIBAPS. The ethical committees from both hospitals approved the study protocol and sample cession. Informed consent was obtained from all blood donors or from their legal representative.

**Reporting summary**. Further information on research design is available in the Nature Research Reporting Summary linked to this article.

## Data availability

All data generated or analyzed during this study are included in this published article (and its Supplementary Information files). Source data are provided with this paper.

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

## Acknowledgements

We are indebted to the IDIBAPS Biobank and to the Hospital Infantil Sant Joan de Deu per a la investigació Biobank, both integrated in the Spanish National Biobank Network of ISCIII, for the sample and data procurement. This work has been supported by projects PI13/01500, PI16/00774 (to R.Ga. and R.Go.) and PI19/00896 (to R.Ga.) integrated in the Plan Estatal de I+D+I and cofinanced by ISCIII-Subdirección General de Evaluación and Fondo Europeo de Desarrollo Regional (FEDER-A way to build Europe); grant 2014 SGR659 (to R.Go.) from the Generalitat de Catalunya; VIII Ayuda SED a Proyectos de Investigación Básica en Diabetes dirigidos por Jóvenes Investigadores (to R.F.R.) from the Sociedad Española de Diabetes Foundation and by the Cátedra Astra-Zeneca (to R.Go.). This work was supported in part by the Intramural Research Program of the NIDCR, NIH (to M.F.Y.). M.F. has been funded by Fundación DiabetesCERO. CIBERDEM (Centro de Investigación Biomédica en Red de Diabetes y Enfermedades Metabólicas Asociadas) is an initiative of the Instituto de Salud Carlos III.

## Author contributions

R.F.R. conducted all experiments. A.G. provided assistance with mouse experiments and immunohistochemistry assays. Y.E., J.M., B.S., and M.F. provided assistance in several experiments. C.B., M.A., and A.W. performed isolation of human islets. M.F.Y. and V.K. provided reagents and performed experiments in bone. R.F.R. and R.Ga. conceived the project. R.F.R., J.V., R.Go., and R.Ga. analyzed and discussed the data. R.F.R. and R.Ga. wrote the paper.

## Competing interests

The authors declare no competing interests.
