## [Peer Review File · Nature Communications]

Reviewers' comments:

Reviewer #1 (Remarks to the Author):

Comment for the authors

In this manuscript, Fernandez-Ruiz and colleagues reported identifying a putative circulating factor, Wisp1, as an inducer of human beta-cell proliferation. Wisp1 is a protein belonging to the CCN protein family, a group of matricellular proteins that regulate the availability of growth factors, involved in different cellular processes as bone development, cell proliferation, production of extracellular matrix and cell migration via regulating the Wnt/beta-catenin pathways. Using antibody arrays, the authors showed a ~5-fold increase in Wisp1 serum levels in mice at 16 post-natal days (young), characterized by high rates of beta-cell proliferation compared to 20-week-old mice (adult). Next, they validated these results using an ELISA and report high Wisp1 expression in bone tissue of young animals but not in adults, and low expression levels in islets compared to other CCN proteins. To assess the specificity of Wisp1 effect, the authors performed eye-chamber transplantation of islets obtained from adult mice into young WT and Wisp1 knock-out mice, reporting low beta-cell proliferation levels in islet transplanted in KO mice. Then, Fernandez-Ruiz et al. overexpressed Wisp1 in liver of 12-week-old mice through adenovirus infections. These mice showed an increase in circulating Wisp1 7 days post-injection, correlating with a 2-fold increase in beta-cell proliferation and 1.5-fold increase in beta-cell mass. However, changes in glucose tolerance were not observed. To evaluate the potential of Wisp1 in reversing hyperglycemia, the authors overexpressed Wisp1 through adenovirus delivery in adult STZ-treated mice and followed them up to 14 days. Despite the increased Wisp1 serum levels and the stimulation of beta-cell proliferation, the effect of Wisp1 was not sufficient to restore normoglycemia in diabetic mice. To explore the mechanism of action of Wisp1 in regulating beta-cell proliferation, the authors exploited co-culture experiments, simultaneously incubating Wisp1-overexpressing NIH-3T3 cells with mouse islets. Expression levels of genes involved in the Wnt/beta-catenin pathways and the cell cycle machinery were found upregulated in islets co-cultured with Wisp1-overexpressing cells for 24/48hr. Since a link between Wisp1 and Akt was previously reported, the authors investigated the effect of Wisp1 on Akt activity in mouse and human islets treated with recombinant Wisp1 protein. Akt phosphorylation levels appeared to be higher in mouse and human islets compared to controls, correlating with higher proliferation rates in beta-cells compared to vehicle-treated samples. Finally, these effects were blunted by using Akt inhibitors in combination with Wisp1 recombinant protein, demonstrating specific Akt effects as a potential mediator of Wisp1 effects on regulating beta-cell proliferation.

Overall the manuscript is reasonable written, however, the vocabulary needs attention. Moreover, important details (e.g. age of mice used in different experimental models, time of incubation of islet/cells) are not provided either in the methods section or in the figure legends. The major concerns are related to poor quality of the pictures showing beta-cell proliferation (low magnification, poor staining quality, Ki67+ cells in insulin-negative areas of islets). There is scant data

from human islets that limits the translational relevance. The authors should address the following major points with additional experiments.

MAJOR POINTS.

1) The authors should expand the Introduction section, discuss relevant published data about role of Wisp1 in liver and bone homeostasis especially in the context of metabolism in humans. This becomes important since the authors argue for the origin of Wisp1 from bone tissue.

2) In Figure 1A, the authors show Ki67 labeling co-staining with insulin as a marker of beta-cell proliferation. Ki67 is a well-established marker of mitosis. However to be certain that the cell cycle is completed in beta-cells and to exclude arrests during the cell cycle phases, the authors should evaluate and include pHH3 labeling levels for all experiments aimed at assessing beta-cell proliferation.

3) Interrogating sera from young versus adult mice using antibody arrays, the authors selected Wisp1 as a candidate for further experiments. However, other soluble proteins also showed remarkable abundance in serum of young mice compared to adult animals (Figure 1C). For example, MCP-1 showed an increase of 100-fold. The author should provide the compelling rationale for focusing on Wisp1 as “the” protein responsible for beta-cell proliferation, considering these other major candidates.

4) In addition, inflammatory cytokines as well as IGF1 or Osteoprotegerin were found up regulated in sera of young mice compared to adults. How can the author exclude the effects (positive or negative) of these proteins in determining the high rates of beta-cell proliferation? Especially since these proteins have been reported to modulate human beta cell proliferation and apoptosis.

5) Despite the specificity of these effects being demonstrated by experiments performed in the Wisp KO mice (Figure 2D), the authors do not report the levels of IGF1 or osteoprotegerin in these animals, leaving open the possibility that these factors are also altered. The authors should evaluate the levels of circulating IGF1, inflammatory cytokines and osteoprotegerin in the sera of these mice to exclude this possibility.

6) In Figure 2D and all the figures showing beta-cell proliferation, the authors report the beta-cell proliferation rates as “fold-change” in comparison to the control samples. This hides the absolute values of basal proliferation in the control group and is unacceptable. The absolute levels should be reported and percentage of Ki67/insulin double + cells can be included as an additional bar graph.

7) Despite the authors use of commercial and validated ELISA kits for assessing the Wisp1 serum levels in mice at different ages, it is necessary to include a positive control next to the mouse serum samples. For example, the authors should spike serum from 20-week-old mice with different amounts of recombinant Wisp1 protein to validate the sensitivity of the ELISA assay and include hem in the data.

8) The authors showed that the Wisp1 gene expression levels are high in the bone tissue of young mice compared to other tissues, concluding that the bone is the main source of soluble Wisp1 (Figure 2C). However, this observation is based only on gene expression data. The authors should evaluate protein levels of Wisp1 in the several tissues being examined in young versus adult mice. In addition, the authors should isolate primary tissues from young vs. adult mice, culture them and collect supernatants for assessing the amounts of secreted Wisp1, to confirm the actual tissue source of soluble Wisp1 in young mice.

9) The authors overexpressed Wisp1 in 12-week-old mice injecting them with adenovirus, resulting in increased Wisp1 expression in the liver and the serum (Figure 3B-C). However, the rationale of using mice at 12-week-old is not clear. The authors should repeat these experiments in 20-week-old animals, as used for the other experiments. An additional confusing issue is this over-expression is likely in liver tissue. While the authors argue that Wisp1 is coming from bone. So this increase by adenovirus expression is targeting a different tissue (liver versus bone).

10) In addition, the authors should consider the opportunity of using the Wisp KO animals for these experiments, to prove whether restoring Wisp1 expression would have any effect on beta-cell proliferation.

11) It is hard to believe that endogenous serum and hepatocyte Wisp1 gene expression levels were not determined in 12-week-old animals injected with Ad-betaGal (Figure 3B-C), considering the expression levels of Wisp1 gene in the liver of 20-week-old animals (Figure 2C). The authors should validate these intriguing results measuring protein levels.

12) Similarly, the authors were not able to determine the levels of endogenous circulating Wisp1 levels in mice injected with Ad-betaGal (Figure 3C). These results are not convincing, given the levels of Wisp1 in the serum of adult mice (Figure 2A).

13) In the experimental model including STZ-treated mice, the authors focused on the measurements of random blood glucose and fasting insulin levels (Figure 4C-D). It will be important to include glucose tolerance and in vivo glucose-stimulated insulin secretion in these mice.

14) To unravel the mechanism of action of Wisp1 in regulating beta-cell proliferation, the authors performed co-culture experiments incubating Wisp1-overexpressing NIH-3T3 cells with mouse islets obtained from adult mice. Although the NIH-3T3 cells represent a suitable in vitro system for overexpressing proteins, it is not the most appropriate model for the purpose of these experiments. The authors should replicate these experiments co-culturing adult mouse islets with bone tissue obtained from young and adult mice or, alternatively, incubate adult mouse islets with conditional media collected from bone tissue cultures.

15) The authors explored the Akt phosphorylation levels in mouse islets treated with recombinant Wisp-1, using vehicle-treated islets as a control sample (Figure 5F-H). The significance of this experiment will increase if the authors will use a homologue of Wisp1 which does not induce beta-cell proliferation (e.g. another member of the CNN family) as an additional control.

16) Although the role of Akt in mediating Wisp1 effect on beta-cell proliferation is demonstrated by using Akt inhibitors in addition to Wisp1 recombinant protein (Figure 5H), the approach used by the authors does not clarify if it is a direct or indirect effect. Also, Akt integrates multiple signaling

pathways. The authors should include inhibitors of growth factor signaling pathways or silence such genes to exclude the contribution of other pathways regulating beta-cell proliferation (e.g. insulin/IGF1R signaling).

17) Overall, the amount of data obtained from human islets is scant. The authors should assess the human beta-cell proliferation *in vivo*, transplanting human islets in young and adult immunodeficient mice. These data will increase the translational impact of this work.

Reviewer #2 (Remarks to the Author):

This is an interesting manuscript, which identifies Wisp1 as a novel circulating factor that promotes beta cell proliferation. The study which is focused on murine tissue only, is well prepared and uses several state of the art *in vivo* and *in vitro* methodology. The following concerns need to be addressed:

1) Please clarify why WISP1 was chosen for further investigation given that the data provided in figure 1 depict other potential circulating factors that are even more increased. Within the array, are proteins detected that have been already described as proliferative/regenerative factors?

2) it would greatly improve the ms if additional translational experiments could be performed to demonstrate that WISP1 is also a potential factor in humans.

3) The mechanisms of WISP1 action, namely Akt-mediated induced proliferation, is well known. Did the authors attempt to expand on this mechanism in more detail? Do the beta cell overexpress a particular integrin that allows them to respond to WISP1? Currently, these data are largely *in vitro* (and only with the recombinant protein), *in vivo* evidence in the models used should be provided and further delineation of the mechanisms beyond Akt phosphorylation would significantly improve the ms.

4) Wisp1 has two known transcripts, which have been reported to result in different proteins that differently effect proliferation. In the Ad Overexpression, which transcript has been used?

5) Several figures missing statistical analysis, which makes proper evaluation of the proposed effect difficult (such as Figure 4 C and D)

6) Western Blots in Figure 5 and 6 are not state of the art. Are these cut? Please provide full Western Blots.

NCOMMS-19-048401

RESPONSES TO REVIEWERS

Reviewer #1:

Overall the manuscript is reasonable written, however, the vocabulary needs attention. Moreover, important details (e.g. age of mice used in different experimental models, time of incubation of islet/cells) are not provided either in the methods section or in the figure legends. The major concerns are related to poor quality of the pictures showing beta-cell proliferation (low magnification, poor staining quality, Ki67+ cells in insulin-negative areas of islets). There is scant data from human islets that limits the translational relevance. The authors should address the following major points with additional experiments.

Authors:

Following the reviewer's concerns we have carefully revised the writing, added detailed information in Methods and included better quality images in many figures. We have also performed additional experiments using human islets to increase the translational relevance of our findings.

Major points

1) The authors should expand the Introduction section, discuss relevant published data about role of Wisp1 in liver and bone homeostasis especially in the context of metabolism in humans. This becomes important since the authors argue for the origin of Wisp1 from bone tissue.

Authors:

We discuss prior literature on the role of Wisp1 in bone homeostasis and human metabolism in the Introduction section of the revised manuscript. Newly added text is highlighted in yellow.

2) In Figure 1A, the authors show Ki67 labeling co-staining with insulin as a marker of beta-cell proliferation. Ki67 is a well-established marker of mitosis. However to be certain that the cell cycle is completed in beta-cells and to exclude arrests during the cell cycle phases, the authors should evaluate and include pHH3 labeling levels for all experiments aimed at assessing beta-cell proliferation.

Authors:

We provide phospho-histone H3 stainings for experiments aimed at assessing beta cell proliferation *in vivo*, including transplanted mouse and human islets (Figures 1 and 3F) and fixed pancreatic tissue from Wisp1 KO and control mice (Figure 3A,B).

3) Interrogating sera from young versus adult mice using antibody arrays, the authors selected Wisp1 as a candidate for further experiments. However, other soluble proteins also showed remarkable abundance in serum of young mice compared to adult animals (Figure 1C). For example, MCP-1 showed an increase of 100-fold. The author should provide the compelling rationale for focusing on Wisp1 as “the” protein responsible for beta-cell proliferation, considering these other major candidates.

4) In addition, inflammatory cytokines as well as IGF1 or Osteoprotegerin were found up regulated in sera of young mice compared to adults. How can the author exclude the effects (positive or negative) of these proteins in determining the high rates of beta-cell proliferation? Especially since these proteins have been reported to modulate human beta cell proliferation and apoptosis.

Authors:

We agree that Wisp1 is unlikely to be the sole responsible factor for beta cell proliferation in lactating mice. In fact, we found, in addition to Wisp1, other proteins enriched in sera from young versus adult mice, some of them with higher order of magnitude than Wisp1 (Figure S2). Moreover, p14 Wisp1 KO pups exhibited a reduction but not a complete impairment in beta cell proliferation (Figures 3A&B). Hence, we have carefully revised the text of the manuscript to make sure that it does not mislead the reader about this point.

On other other hand, we chose Wisp1 based on the following points:

- Our focus was the identification of extrinsic regulators of beta cell proliferation present in young blood. We specifically searched for factors not produced by islets. In this regard, we validated that Wisp1 was not endogenously expressed in islets by qRT-PCR (Figure 2D&E). By contrast, prior literature and unpublished data from our group revealed endogenous production by beta cells of some of the other factors identified in the arrays. For example: MCP-1 is secreted by rodent and human beta cells (PLoS ONE 7(10): e46986, 2012; Diabetes 51: 55-65, 2002); osteoprotegerin is endogenously enhanced in models of beta cell expansion (Cell Metab 22:77-85, 2015) and osteopontin is induced by GIP in human islets (Diabetes 60: 2424-33, 2011). Further, we found that Igf1 and osteopontin are both highly expressed in islets from lactating mice (JMir, RGasa, unpublished data).
- Wisp1 is associated with obesity and insulin resistance in humans, a condition characterized by beta cell mass expansion.
- The involvement of Wisp1 in bone homeostasis and the fact that bone is known to secrete proteins that affect pancreatic beta cells including osteocalcin (Diabetes 63:1021-1031, 2014), osteopontin (Endocrinology 148:575-584, 2007) and osteoprotegerin (Cell Metab 22:77-85, 2015).

5) Despite the specificity of these effects being demonstrated by experiments performed in the Wisp KO mice (Figure 2D), the authors do not report the levels of IGF1 or osteoprotegerin in these animals, leaving open the possibility that these factors are also

altered. The authors should evaluate the levels of circulating IGF1, inflammatory cytokines and osteoprotegerin in the sera of these mice to exclude this possibility.

Authors:

We have measured circulating levels of Igf1, osteoprotegerin (OPG) and osteopontin in p14 Wisp1 KO and their WT littermates using commercially available ELISAs. Wisp1 KO exhibit similar Igf1 and osteopontin and modestly decreased OPG serum levels as compared to WT littermates (new Figure S4). Despite we cannot completely rule out an impact of lower OPG on beta cell proliferation in Wisp1 KO mice, the degree of reduction (17%) is so small that we find it improbable that it is the principal cause of the nearly 50% decline in beta cell proliferation in this model. Interestingly, we have also measured serum OPG levels in C57B6 mice at different ages and confirmed an age-dependent decline in serum levels of this protein (shown in the graph on the right). Of note, whilst serum OPG levels are similar between p14 and p28, beta cell proliferation rapidly declines between these two postnatal ages (Pardo et al. *Diabetologia* **55**,3331–3340;2012), thus pointing to a minor role of circulating OPG in the early postnatal down-regulation of beta cell proliferative activity.

Lastly, we have also determined serum levels of Igf1, osteoprotegerin and osteopontin in p14 Wisp1 KO mice treated with recombinant Wisp1 protein for three days and have found no differences when compared with untreated p14 Wisp1 KO littermates (new Figure S4).

6) In Figure 2D and all the figures showing beta-cell proliferation, the authors report the beta-cell proliferation rates as “fold-change” in comparison to the control samples. This hides the absolute values of basal proliferation in the control group and is unacceptable. The absolute levels should be reported and percentage of Ki67/insulin double + cells can be included as an additional bar graph.

Authors:

Following the reviewer’s concern we now provide percentages (%) of double positive Insulin/Ki67 and Insulin/pHH3 relative to total beta (insulin+) cells throughout the manuscript.

7) Despite the authors’ use of commercial and validated ELISA kits for assessing the Wisp1 serum levels in mice at different ages, it is necessary to include a positive control next to the mouse serum samples. For example, the authors should spike serum from 20-week-old mice with different amounts of recombinant Wisp1 protein to validate the sensitivity of the ELISA assay and include them in the data.

Authors:

We have validated the mouse Wisp1 ELISA by determining Wisp1 concentrations in culture media 48 hours after the addition of different amounts of recombinant mouse Wisp1 protein. Results are shown in the graph on the right.

We don't find it necessary to include these validation data in the manuscript but are willing to do it upon the reviewer's request.

8) The authors showed that the Wisp1 gene expression levels are high in the bone tissue of young mice compared to other tissues, concluding that the bone is the main source of soluble Wisp1 (Figure 2C). However, this observation is based only on gene expression data. The authors should evaluate protein levels of Wisp1 in the several tissues being examined in young versus adult mice. In addition, the authors should isolate primary tissues from young vs. adult mice, culture them and collect supernatants for assessing the amounts of secreted Wisp1, to confirm the actual tissue source of soluble Wisp1 in young mice.

Authors:

To address the issue raised by the reviewer we have performed additional experiments using bone tissue from lactating and adult mice. Thus, Figure S3A-B includes images of standard IHC of bone tissue stained for Wisp1 (antibody described in PlosOne 8: e71709, 2013), which reveal enhanced Wisp1 staining in young as compared to adult bone. On the other hand, Figure S3C presents the concentration (measured by ELISA) of Wisp1 secreted by isolated cells from calvaria (skull) bone from young and adult mice. The graph shows that Wisp1/total protein secreted is significantly higher in bone cell cultures from young relative to adult mice. These new experiments together with the gene expression assay shown in Figure 2F support the idea that the bone is the likely source of circulating Wisp1 in lactating mice.

9) The authors overexpressed Wisp1 in 12-week-old mice injecting them with adenovirus, resulting in increased Wisp1 expression in the liver and the serum (Figure 3B-C). However, the rationale of using mice at 12-week-old is not clear. The authors should repeat these experiments in 20-week-old animals, as used for the other experiments.

Authors:

Mice with ages ranging from 3 to 6 months (12 to 24 weeks) are commonly considered mature adult. We have used indistinctively mice within this age range to perform our experiments because they display similar low levels of circulating Wisp1 (please note that we include data on serum Wisp1 levels in 11 week-old mice in Figure 2B of our revised submission). Additionally, mice between 11 to 24 weeks old present similar beta cell proliferation rates (see graph below, data from our laboratory in the frame of a separate study which is in accordance with literature in the field).

An additional confusing issue is this over-expression is likely in liver tissue. While the authors argue that Wisp1 is coming from bone. So this increase by adenovirus expression is targeting a different tissue (liver versus bone).

Authors:

The aim of the *in vivo* adenovirus experiments was to measure endogenous pancreatic beta cell proliferation upon increasing serum levels of Wisp1 in adult mice. Targeting the liver via tail vein injection of viral vectors or DNA plasmids is broadly used to achieve meaningful systemic levels of many proteins (some examples: PNAS 93: 14804-14808, 1996; JBC 276: 6343-6449, 2001; Human Mol Gen 11:43-58, 2002; Cell 159:691-696, 2014). We chose this approach over multiple injections of recombinant Wisp1 protein because we considered that it would permit steady-state availability of Wisp1 in a cost-wise and reliable manner in adult animals. We considered that the tissue source for Wisp1 in these over-expression experiments was not a central issue, as long as we were able to validate increased serum levels of this factor (Figures 4C and 5C).

10) In addition, the authors should consider the opportunity of using the Wisp KO animals for these experiments, to prove whether restoring Wisp1 expression would have any effect on beta-cell proliferation.

Authors:

We thank the reviewer for suggesting this experiment. Having demonstrated that p14 Wisp1 KO pups exhibited lower beta cell proliferation relative to their littermate controls (Figure 3A&B), we queried whether restoring circulating Wisp1 would have a positive impact on beta cell proliferation in Wisp1 KO pups. To do this experiment we performed daily injections of recombinant Wisp1 protein to p9 Wisp1 KO pups (adenoviral infusion was not possible due to small size of the animals) for three days and harvested the pancreas at the end of this period for immuno-staining analysis. As shown in new Figures 3C and 3D we found a significant increase in beta cell proliferation in pups treated with recombinant Wisp1, further proving a positive role of this factor in postnatal beta cell growth.

11) It is hard to believe that endogenous serum and hepatocyte Wisp1 gene expression levels were not determined in 12-week-old animals injected with Ad-betaGal (Figure 3B-

C), considering the expression levels of *Wisp1* gene in the liver of 20-week-old animals (Figure 2C). The authors should validate these intriguing results measuring protein levels. 12) Similarly, the authors were not able to determine the levels of endogenous circulating *Wisp1* levels in mice injected with Ad-betaGal (Figure 3C). These results are not convincing, given the levels of *Wisp1* in the serum of adult mice (Figure 2A).

Authors:

To address this point, we have measured mouse *Wisp1* mRNA levels in livers from animals injected with recombinant adenoviruses encoding WISP1 and betaGal, both under physiological (Figure 4B) and diabetic conditions (Figure 5B). Results are included in new Figure 4B and Figure 5B and reveal that endogenous *Wisp1* mRNA levels are not modified in response to human WISP1 overexpression in the liver.

In addition, we measured mouse *Wisp1* levels in archived sera from mice injected with AdBetaGal or Ad-WISP1, under healthy and diabetic conditions. As shown in the graph on the right, corresponding to day 4 post-infection, there were no changes in circulating mouse *Wisp1* levels among the different treatment groups. These data are not included in the revised manuscript, but are willing to do so as Supplementary information upon the reviewer's request.

13) In the experimental model including STZ-treated mice, the authors focused on the measurements of random blood glucose and fasting insulin levels (Figure 4C-D). It will be important to include glucose tolerance and in vivo glucose-stimulated insulin secretion in these mice.

Authors:

We have performed intraperitoneal glucose tolerance and glucose-induced insulin secretion tests in a new cohort of animals. As shown in Figure S5, overexpression of WISP1 had no measurable effect in neither test. Remarkably, differences in fed insulinemia between Ad.Wisp1 and Ad-betaGal-injected mice have become statistically significant after increasing the number of mice in both treatment groups (Figure 5E, $p < 0.01$).

14) To unravel the mechanism of action of *Wisp1* in regulating beta-cell proliferation, the authors performed co-culture experiments incubating *Wisp1*-overexpressing NIH-3T3 cells with mouse islets obtained from adult mice. Although the NIH-3T3 cells represent a suitable in vitro system for overexpressing proteins, it is not the most appropriate model for the purpose of these experiments. The authors should replicate these experiments co-culturing adult mouse islets with bone tissue obtained from young and adult mice or,

alternatively, incubate adult mouse islets with conditional media collected from bone tissue cultures.

Authors:

To conclusively validate the beta cell mitogenic role of Wisp1 *in vitro*, incubation of mouse islets with bone tissue from different ages (or alternatively, their conditioned media) has an important limitation, which is that the bone secretes other factors that, like Wisp1, can stimulate beta cell proliferation and whose levels may also vary with age. Hence, it would be challenging to discriminate the effects of Wisp1 from the other bone-borne factors.

As an alternative, we have used two different models: treatment with recombinant proteins and co-culture with Wisp1-overexpressing 3T3 cells. The rationale for including the latter model was to test a model system where Wisp1 levels would be closer to physiological concentrations of this protein. In addition, to the best of our knowledge, fibroblast cells under basal conditions do not promote beta cell proliferation. Please note that we have reasoned the use of the 3T3 experimental model in the revised manuscript, page 15, in yellow. We are confident that these two approaches are sufficient to definitely prove the direct involvement of Wisp1 in beta cell proliferation.

15) The authors explored the Akt phosphorylation levels in mouse islets treated with recombinant Wisp-1, using vehicle-treated islets as a control sample (Figure 5F-H). The significance of this experiment will increase if the authors will use a homologue of Wisp1 which does not induce beta-cell proliferation (e.g. another member of the CNN family) as an additional control.

Authors:

The CCN family consists of six proteins including Wisp1 (CCN4). Remarkably, both Cyr61 (CCN1) and CTGF (CCN2) have been shown to promote beta and pancreatic cell proliferation, respectively. However, whether any of the rest (CCN3, CCN5 or CCN6) shares this ability has not been documented. We agree with the reviewer that this could be an interesting experiment. However, while it would improve our comprehension of the similarities/dissimilarities among CCN proteins, it is not indispensable to recognize the participation of AKT in the actions of Wisp1. The involvement of AKT in the proliferative effects of Wisp1 in beta cells is clearly demonstrated by using AKT inhibitors both in mouse and human islets.

16) Although the role of Akt in mediating Wisp1 effect on beta-cell proliferation is demonstrated by using Akt inhibitors in addition to Wisp1 recombinant protein (Figure 5H), the approach used by the authors does not clarify if it is a direct or indirect effect. Also, Akt integrates multiple signaling pathways. The authors should include inhibitors of growth factor signaling pathways or silence such genes to exclude the contribution of other pathways regulating beta-cell proliferation (e.g. insulin/IGF1R signaling).

Authors:

We believe that our findings showing complete impairment of Wisp1-induced beta cell proliferation in the presence of the Akt chemical inhibitors support the conclusion that Akt mediates the pro-proliferative actions of Wisp1 in mouse and human islets. Furthermore, the positive role of Akt in beta cell proliferation has been previously established (Nat Med 7:1133-1137, 2001; J Clin Invest 108:1631-1638, 2001).

Whether Wisp1 might synergize with other growth factors, via Akt or alternative intracellular mediators, is indeed a stimulating topic that deserves to be evaluated in the future. However, given the time and resources needed to perform these studies in a comprehensive manner, we feel that they fall beyond the scope of the present manuscript.

Nonetheless, we considered relevant to compare the effects of Wisp1 with other known beta cell mitogens. Specifically, we have tested the effects of Harmine, a DYRK1a inhibitor, both in mouse and human islets. Interestingly, Harmine is more potent than Wisp1 in mouse islets, whilst it has similar potency in human islets. These data are shown in Figure 6 of the revised manuscript.

17) Overall, the amount of data obtained from human islets is scant. The authors should assess the human beta-cell proliferation *in vivo*, transplanting human islets in young and adult immunodeficient mice. These data will increase the translational impact of this work.

Authors:

Following the reviewer's suggestion we have transplanted human islets into the ACE of both lactating and adult immuno-compromised mouse recipients and corroborated that human beta cells proliferate more in a young environment (Figure 1C, D).

In addition we have included new additional information involving human samples:

- 1- Circulating WISP1 levels are higher in children than in adult (Figure 2C)
- 2- Like in mouse islets, WISP1 is not endogenously expressed in human islets. This result validates the role of Wisp1 as an extrinsic factor (Figure 2E).

Reviewer #2 (Remarks to the Author):

This is an interesting manuscript, which identifies Wisp1 as a novel circulating factor that promotes beta cell proliferation. The study which is focused on murine tissue only, is well prepared and uses several state of the art *in vivo* and *in vitro* methodology. The following concerns need to be addressed:

1) Please clarify why WISP1 was chosen for further investigation given that the data provided in figure 1 depict other potential circulating factors that are even more increased. Within the array, are proteins detected that have been already described as proliferative/regenerative factors?

Authors:

(please note that this response is shared with reviewer #1, point #1)

We agree that *Wisp1* is unlikely to be the sole responsible factor for beta cell proliferation in lactating mice. In fact, we found, in addition to *Wisp1*, other proteins enriched in sera from young versus adult mice, some of them with higher order of magnitude than *Wisp1* (Figure S2). Moreover, p14 *Wisp1* KO pups exhibited a reduction but not a complete impairment in beta cell proliferation (Figures 3A&B). Hence, we have carefully revised the text of the manuscript to make sure that it does not mislead the reader about this point.

On other other hand, we chose *Wisp1* based on the following points:

- Our focus was the identification of extrinsic regulators of beta cell proliferation present in young blood. We specifically searched for factors not produced by islets. In this regard, we validated that *Wisp1* was not endogenously expressed in islets by qRT-PCR (Figure 2). By contrast, prior literature and unpublished data from our group revealed endogenous production by beta cells of some of the other factors identified in the arrays. For example: MCP-1 is secreted by rodent and human beta cells (PLoS ONE 7(10): e46986, 2012; Diabetes 51: 55-65, 2002); osteoprotegerin is endogenously enhanced in models of beta cell expansion (Cell Metab 22:77-85, 2015) and osteopontin is induced by GIP in human islets (Diabetes 60: 2424-33, 2011). Further, we found that *Igf1* and osteopontin are both highly expressed in islets from lactating mice (JMir, RGasa, unpublished data).
- *Wisp1* is associated with obesity and insulin resistance in humans, a condition characterized by beta cell mass expansion.
- The involvement of *Wisp1* in bone homeostasis and the fact that bone is known to secrete proteins that affect pancreatic beta cells including osteocalcin (Diabetes 63:1021-1031, 2014), osteopontin (Endocrinology 148:575-584, 2007) and osteoprotegerin (Cell Metab 22:77-85, 2015).

2) It would greatly improve the ms if additional translational experiments could be performed to demonstrate that *WISP1* is also a potential factor in humans.

(please note that this response is shared with reviewer #1, point #17)

Authors:

Following the reviewer's suggestion we have transplanted human islets into the ACE of both lactating and adult immuno-compromised mouse recipients and corroborated that human beta cells proliferate more in a young environment (Figure 1C, D).

In addition we have include new additional information involving human samples:

- 1- Circulating *WISP1* levels are higher in children than in adult sera (Figure 2C)
- 2- *WISP1* is not endogenously expressed in human islets, validating the role of *Wisp1* as an extrinsic factor (Figure 2E).

3) The mechanisms of WISP1 action, namely Akt-mediated induced proliferation, is well known. Did the authors attempt to expand on this mechanism in more detail? Do the beta cell overexpress a particular integrin that allows them to respond to WISP1?

Authors:

Unraveling the mechanism whereby Wisp1 activates Akt in islets is a relevant question that we also wish to answer. Unfortunately, it is not a straightforward task that can be properly addressed in a timely manner in the frame of this revision. In fact, available literature on the mode of action of Wisp1 in other cellular contexts is also rather limited. Experiments to determine the effect of blocking integrin signaling or synergism of Wisp1 with other signaling factors are on their way and, if successful, will help us answer this question. However, we feel that these experiments are beyond the scope of the present study.

Currently, these data are largely *in vitro* (and only with the recombinant protein), *in vivo* evidence in the models used should be provided and further delineation of the mechanisms beyond Akt phosphorylation would significantly improve the ms.

Authors:

Following the reviewer's request we have studied Akt in islets isolated from p14 Wisp1 WT and KO mice. As shown in the figure below, we have found no differences in the activation status or total protein levels of this kinase between KO and WT islets. Whilst this observation is apparently at odds with our conclusion that Akt mediates Wisp1 effects in islets, it should be noted that acute stimulation and steady state activity of Akt (or any other kinase) might diverge. Acute stimulation experiments are conducted in cells under starvation conditions (reduced or absent serum) in order to reduce background and allow detection of changes in Akt activation upon addition of exogenous factors. However, this cannot be easily controlled *in vivo*. In fact, Akt is at the crossroads of multiple pathways and adaptation and feedback regulatory mechanism likely take place *in vivo*. We haven't included this result in the revised manuscript because we feel it doesn't validate nor invalidate that Akt is downstream of Wisp1 in islets. However, we are willing to include it upon the reviewer's request.

4) Wisp1 has two known transcripts, which have been reported to result in different proteins that differently effect proliferation. In the Ad Overexpression, which transcript has been used?

Authors:

The adenovirus encodes the full-length WISP1 cDNA. The cDNA construct was obtained from the Levine lab (Genes Dev 16:46-57, 2002). Generation of the adenovirus used in our study is described in Ono et al. J Bone Miner Res, 26:193-208 (2011).

5) Several figures missing statistical analysis, which makes proper evaluation of the proposed effect difficult (such as Figure 4 C and D)

Authors:

We have revised all figures and included statistical significance if applicable. Note that differences in blood glucose levels in Figure 4C (new Figure 5D) remain not statistically significant. Conversely, differences in serum insulin, which were nearly significant ($p=0.07$) in original Figure 4D have reached statistical significance ($p < 0.01$) after increasing the sample number during revision of the manuscript (new Figure 5E).

6) Western Blots in Figure 5 and 6 are not state of the art. Are these cut? Please provide full Western Blots.

Authors:

We have included new immunoblot images. Please note that the manuscript has been re-organized and previous Figure 5G is now Figure 7A and previous Figure 6B is now Figure 7B. Image in Figure 7A is shown uncut, whereas image in Figure 7B has been cut (this is marked with a dotted line in the figure). Below we provide the full western blot image for the reviewer.

Human islets: Figure 7c

REVIEWER COMMENTS

Reviewer #1 (Remarks to the Author):

The authors have made a good faith effort to respond to the critiques. This has strengthened the manuscript. There are a couple issues listed below that still need attention.

1. In regard to point #2. The pHH3 data are OK. However the quality of pictures of the insulin/Ki67 and insulin/pHH3 co-staining to show double positive cells is unacceptable. The authors should provide single-channel figures and higher magnification for each. It is impossible to appreciate the proliferating beta-cells with the current pictures all through the manuscript. This is critical for the readers and publication.

2. In read to point # 10: The authors response is OK. However, authors should provide the serum levels of the Wisp KO mice treated with recombinant Wisp. This will provide evidence if they are similar to wild type animals, considering that beta-cell proliferation levels are almost 2-fold higher (yellow bar in Figure 3a vs. Orange bar in figure 3c). It is important to note that mice used in figure 3a-b were p14 and mice used in 3c-d were p12; this is relevant because it is well known that beta-cell proliferation levels drop down dramatically at early stages. A comment should be made in this regard with references.

NCOMMS-19-048401-R1

RESPONSES TO REVIEWERS

Reviewer #1:

The authors have made a good faith effort to respond to the critiques. This has strengthened the manuscript. There are a couple issues listed below that still need attention.

1. In regard to point #2. The pHH3 data are OK. However the quality of pictures of the insulin/Ki67 and insulin/pHH3 co-staining to show double positive cells is unacceptable. The authors should provide single-channel figures and higher magnification for each. It is impossible to appreciate the proliferating beta-cells with the current pictures all through the manuscript. This is critical for the readers and publication.

Authors:

We provide better quality images throughout the manuscript. In addition, for some of the Figures we include single-channel images and image amplifications to help better appreciate proliferating beta cells.

2. In read to point # 10 (in vivo recovery of KO with rmWisp1): The authors response is OK. However, authors should provide the serum levels of the Wisp KO mice treated with recombinant Wisp (A). This will provide evidence if they are similar to wild type animals, considering that beta-cell proliferation levels are almost 2-fold higher (yellow bar in Figure 3a vs. Orange bar in figure 3c).

Authors:

Following the reviewer's comment, we have determined mouse Wisp1 levels by ELISA in serum samples from p12 Wisp KO mice treated with saline or with recombinant mouse Wisp1 (rmWisp1) at 1mg/Kg body weight (3.4 to 6 ug per injection) during three days (p9 to p11). The dose of 1mg/Kg was chosen based on existing publications using intraperitoneal (IP) injections of Wisp1 in mice (Lutjanenko L. et al. Cell Stem Cell 24, 433-446, 2019; Tong Y. et al. Sci Rep 6:20141, 2016).

As shown in the graph below, Wisp1 protein is detected in the serum of Wisp1KO pups injected with rmWisp1 but not in KO pups treated with saline.

However, serum Wisp1 levels are substantially lower in rmWisp1-treated Wisp1KO pups than in p14 control mice (11.6 ng/ml, shown in Figure 2b manuscript). This result was not unexpected as IP administration typically results in low systemic concentrations as compared, for example, to intravenous routes. Yet, taken into consideration the small size of the animals and the fact that the dam can reject or cannibalize neonates that have been handled, we opted for IP administration because of its simplicity and fastness. Moreover, IP injections have the advantages that (i) they can deliver high doses of administered drugs in close proximity to intraperitoneal organs including the pancreas, and (ii) they allow quite long periods of absorption from the repository site (Routes of administration, The laboratory mouse, ISBN:978-0-12-382008-2). We think that these two features have been instrumental to allow exposure of the pancreas to relevant concentrations of rmWisp1 in Wisp1KO pups. Taken together, these considerations question the relevance of serum Wisp1 levels in our experiment and hence have opted not to include these data in the revised manuscript. However, we are willing to do so upon the reviewer's request.

Nonetheless, the reviewer's comment has helped us realize that the sentence "to examine whether *restitution of Wisp1 in the circulation*" that we had used to introduce this experiment was inappropriate. We have changed it for "we determined if administration of recombinant mouse Wisp1 to Wisp1^{-/-} animals could enhance postnatal beta cell proliferation in vivo" and re-written the paragraph hereafter.

It is important to note that mice used in figure 3a-b were p14 and mice used in 3c-d were p12; this is relevant because it is well known that beta-cell proliferation levels drop down dramatically at early stages. A comment should be made in this regard with references

Due to the rapid decline in beta cell replication during the neonatal period (see, as an example, graph on the right depicting unpublished results by our group from C57B6 mice), it is predictable that

beta cell proliferation rates decrease to some extent between postnatal days 12 and p14. As suggested by the reviewer, we include a comment and a reference on this point in the revised manuscript. Yet, we believe that the magnitude of the differences in proliferation rates between Figures 3a,b and Figures 3c,d may have also been influenced by the known variability associated to the immunofluorescence staining technique. Note that both experiments were conducted more than a year apart and reagents (in particular, antibody lots) and some of the equipment used was not the same. In any case, for each experiment, controls were always performed in parallel, so that we are confident of the differences observed between experimental groups.

Please note that Reviewer 2 was not available to reassess your paper at this time. Given the nature of Reviewer 2's comments we have decided to not seek further advice on the replies to their report from a substitute referee. However, we would like to ask you to please clarify in a revised manuscript if WISP detected in circulation is the same isoform as the one used for overexpression.

Authors:

We have clarified in the Methods section that the ELISAs used to detect mouse and human WISP1 in circulation detect the full-length proteins. Recombinant proteins and adenovirally-expressed WISP1 used in our study are also full-length versions of Wisp1.

REVIEWER COMMENTS

Reviewer #1 (Remarks to the Author):

Although the effort by the authors in providing clearer pictures has helped somewhat in appreciating proliferating beta-cells, several of the figures are still non convincing.

Specifically:

- Figure 3d, panel representing merged staining of *Wisp1*^{-/-} + *mrWisp1* samples: it is unclear whether the pHH3 staining merges with the Hoescht staining. The authors should provide clearer pictures.
- Figure 4d, panel representing merged staining of Ad-betaGal samples: the reported beta-cell proliferation levels in the quantification plot (Figure 4e) are ~0.5%, therefore the authors should select a representative picture that indicates these results.
- Figure 4d, panel representing merged staining of Ad-WISP1 samples: the selected proliferative cells are in an area where insulin staining is null/very faint, so it is unclear whether these are actual beta-cells. The authors should provide pictures with adequate and clear insulin staining.
- Figure 6a, panel representing merged staining of *rmWisp1*-250 samples: the selected proliferative cells are in an area where insulin staining is null/very faint, so it is unclear whether these are real beta-cells. The authors should provide pictures with adequate insulin staining.
- Figure 6a, panel representing merged staining of control samples: the reported beta-cell proliferation levels in the quantification plot (Figure 6b) are ~0.5%, therefore the authors should select a representative picture that indicates these results.
- Figure 6c, panel representing merged staining of Ad-betaGal 48hr samples: the reported beta-cell proliferation levels in the quantification plot (Figure 6d) were similar to those in Ad-WISP1 24 hr samples, therefore the authors should select a representative picture that indicates these results.
- Figure 6e, panel representing merged staining of *rhWISP1*-500 samples: the selected proliferative cells are in an area where insulin staining is null/very faint, so it is unclear whether these are beta-cells. The authors should provide pictures with adequate insulin staining.
- Figure 7b, panel representing merged staining of control and AZD samples: the reported beta-cell proliferation levels in the quantification plot (Figure 7c) are ~0.5%, therefore the authors should select a representative picture that indicates these results.
- Figure 7e, panel representing merged staining of control and AZD samples: the reported beta-cell proliferation levels in the quantification plot (Figure 7c) are ~1% and ~0.5% respectively, therefore the authors should select a representative picture that supports these results.

RESPONSE TO REVIEWERS

Reviewer #1:

Although the effort by the authors in providing clearer pictures has helped somewhat in appreciating proliferating beta-cells, several of the figures are still non convincing.

Specifically:

- Figure 3d, panel representing merged staining of Wisp1^{-/-} + mrWisp1 samples: it is unclear whether the pHH3 staining merges with the Hoescht staining. The authors should provide clearer pictures.
- Figure 4d, panel representing merged staining of Ad-betaGal samples: the reported beta-cell proliferation levels in the quantification plot (Figure 4e) are ~0.5%, therefore the authors should select a representative picture that indicates these results.
- Figure 4d, panel representing merged staining of Ad-WISP1 samples: the selected proliferative cells are in an area where insulin staining is null/very faint, so it is unclear whether these are actual beta-cells. The authors should provide pictures with adequate and clear insulin staining.
- Figure 6a, panel representing merged staining of rmWisp1-250 samples: the selected proliferative cells are in an area where insulin staining is null/very faint, so it is unclear whether these are real beta-cells. The authors should provide pictures with adequate insulin staining.
- Figure 6a, panel representing merged staining of control samples: the reported beta-cell proliferation levels in the quantification plot (Figure 6b) are ~0.5%, therefore the authors should select a representative picture that indicates these results.
- Figure 6c, panel representing merged staining of Ad-betaGal 48hr samples: the reported beta-cell proliferation levels in the quantification plot (Figure 6d) were similar to those in Ad-WISP1 24 hr samples, therefore the authors should select a representative picture that indicates these results.
- Figure 6e, panel representing merged staining of rhWISP1-500 samples: the selected proliferative cells are in an area where insulin staining is null/very faint, so it is unclear whether these are beta-cells. The authors should provide pictures with adequate insulin staining.
- Figure 7b, panel representing merged staining of control and AZD samples: the reported beta-cell proliferation levels in the quantification plot (Figure 7c) are ~0.5%, therefore the authors should select a representative picture that indicates these results.
- Figure 7e, panel representing merged staining of control and AZD samples: the reported beta-cell proliferation levels in the quantification plot (Figure 7c) are ~1% and ~0.5% respectively, therefore the authors should select a representative picture that supports these results.

Authors:

We have revised all the figure panels indicated by the reviewer and selected new images to provide more representative pictures supporting the results shown in the bar graphs (Fig 4d; Fig 6a/c; Fig 7 b/e). We have also provided images with better insulin staining to improve the identification of beta cells (Fig 4d; Fig 6e). Lastly, we include in Fig. 3d a clearer image

showing pHH3/Hoescht co-staining. We hope that the reviewer finds these changes fitting. We want to thank the reviewer for his/her precise and meticulous revision that has undoubtedly helped to improve our manuscript.

REVIEWERS' COMMENTS

Reviewer #1 (Remarks to the Author):

The authors have responded to the critiques.